# Nearly Optimal Algorithms for Contextual Dueling Bandits from Adversarial Feedback

## Abstract

Learning from human feedback plays an important role in aligning generative models, such as large language models (LLM). However, the effectiveness of this approach can be influenced by adversaries, who may intentionally provide misleading preferences to manipulate the output in an undesirable or harmful direction. To tackle this challenge, we study a specific model within this problem domain–contextual dueling bandits with adversarial feedback, where the true preference label can be flipped by an adversary. We propose an algorithm namely robust contextual dueling bandits (RCDB), which is based on uncertainty-weighted maximum likelihood estimation. Our algorithm achieves an $\widetilde{O}(d\sqrt{T} + dC)$ regret bound, where $T$ is the number of rounds, $d$ is the dimension of the context, and $0 \leq C \leq T$ is the total number of adversarial feedback. We also prove a lower bound to show that our regret bound is nearly optimal, both in scenarios with and without ($C = 0$) adversarial feedback. To the best of our knowledge, our work is the first to achieve nearly minimax optimal regret for dueling bandits in the presence of adversarial preference feedback. Additionally, we conduct experiments to evaluate our proposed algorithm against various types of adversarial feedback. Experimental results demonstrate its superiority over the state-of-the-art dueling bandit algorithms in the presence of adversarial feedback.

## 1 Introduction

Acquiring an appropriate reward proves challenging in numerous real-world applications, often necessitating intricate instrumentation (Zhu et al., 2020) and time-consuming calibration (Yu et al., 2020) to achieve satisfactory levels of sample efficiency. For instance, in training large language models (LLM) using reinforcement learning from human feedback (RLHF), the diverse values and perspectives of humans can lead to uncalibrated and noisy rewards (Ouyang et al., 2022). In contrast, preference-based data, which involves comparing or ranking various actions, is a more straightforward method for capturing human judgments and decisions. In this context, the dueling bandit model (Yue et al., 2012) provides a problem framework that focuses on optimal decision-making through pairwise comparisons, rather than relying on the absolute reward for each action.

However, human feedback may not always be reliable. In real-world applications, human feedback is particularly vulnerable to manipulation through preference label flip. Adversarial feedback can significantly increase the risk of misleading a large language model (LLM) into erroneously prioritizing harmful content, under the false belief that it reflects human preference. Despite the significant influence of adversarial feedback, there is limited existing research on the impact of adversarial feedback specifically within the context of dueling bandits. A notable exception is Agarwal et al. (2021), which studies dueling bandits when an adversary can flip some of the preference labels received by the learner. They proposed an algorithm that is agnostic to the amount of adversarial feedback introduced by the adversary. However, their setting has the following two limitations. First, their study was confined to a finite-armed setting, which renders their results less applicable to modern applications such as RLHF. Second, their adversarial feedback is defined on the whole comparison matrix. In each round, the adversary observes the outcomes of all pairwise comparisons and then decides to corrupt some of the pairs before the agent selects the actions. This assumption does not align well with the real-world scenario, where the adversary often flips the preference label based on the information of the selected actions.

In this paper, to address the above challenge, we aim to develop contextual dueling bandit algorithms that are robust to adversarial feedback. This enables us to effectively tackle problems involving a large number of actions while also taking advantage of contextual information. We specifically

Table 1: Comparison of algorithms for robust bandits and dueling bandits.

| Model | Algorithm | Setting | Regret |
|---|---|---|---|
| Bandits | Multi-layer Active Arm Elimination Race (Lykouris et al., 2018) | $K$-armed Bandits | $\widetilde{O}(K^{1.5}C\sqrt{T})$ |
| | BARBAR (Gupta et al., 2019) | $K$-armed Bandits | $\widetilde{O}(\sqrt{KT} + KC)$ |
| | SBE (Li et al., 2019) | Linear Bandits | $\widetilde{O}(d^2C/\Delta + d^5/\Delta^2)$ |
| | Robust Phased Elimination (Bogunovic et al., 2021) | Linear Bandits | $\widetilde{O}(\sqrt{dT} + d^{1.5}C + C^2)$ |
| | Robust weighted OFUL (Zhao et al., 2021) | Linear Contextual Bandits | $\widetilde{O}(dC\sqrt{T})$ |
| | CW-OFUL (He et al., 2022) | Linear Contextual Bandits | $\widetilde{O}(d\sqrt{T} + dC)$ |
| Dueling Bandits | WIWR (Agarwal et al., 2021) | $K$-armed Dueling Bandits | $\widetilde{O}(K^2C/\Delta_{\min} + \sum_{i \neq i^\star} K^2/\Delta_i^2)$ |
| | Versatile-DB (Saha & Gaillard, 2022) | $K$-armed Dueling Bandits | $\widetilde{O}(C + \sum_{i \neq i^\star} 1/\Delta_i + \sqrt{K})$ |
| | RCDB (**Our work**) | Contextual Dueling Bandits | $\widetilde{O}(d\sqrt{T} + dC)$ |

consider a scenario where the adversary knows the selected action pair and the true preference of their comparison. In this setting, the adversary's only decision is whether to flip the preference label or not. We highlight our contributions as follows:

- We propose a new algorithm called robust contextual dueling bandits (RCDB), which integrates uncertainty-dependent weights into the Maximum Likelihood Estimator (MLE). Intuitively, our choice of weight is designed to induce a higher degree of skepticism about potentially "untrustworthy" feedback. The agent is encouraged to focus more on feedback that is more likely to be genuine, effectively diminishing the impact of any adversarial feedback.
- We analyze the regret of our algorithm under at most $C$ number of adversarial feedback. For known adversarial level, our result consists of two terms: a $C$-independent term $\widetilde{O}(d\sqrt{T})$, which matches the lower bound established in Bengs et al. (2022) for uncorrupted linear contextual dueling bandits, and a $C$-dependent term $\widetilde{O}(dC)$. Furthermore, we establish a lower bound for dueling bandits with adversarial feedback, demonstrating the optimality of our adversarial term. Consequently, our algorithm for dueling bandits attains the optimal regret in both scenarios, with and without adversarial feedback.
- When the adversarial level is unknown, we conduct our algorithm with an optimistic estimator of the number of adversarial feedback and prove the optimality of our result in case of a strong adversary. To the best of our knowledge, our work is the first to achieve nearly minimax optimal regret for dueling bandits in the presence of adversarial preference feedback, regardless of whether the amount of adversarial feedback is known.
- We conduct extensive experiments to validate the effectiveness of our algorithm RCDB. To comprehensively assess RCDB's robustness against adversarial feedback, we evaluate its performance under various types of adversarial feedback and compare the results with state-of-the-art dueling bandit algorithms. Experimental results demonstrate the superiority of our algorithm in the presence of adversarial feedback, which corroborate our theoretical analysis.

**Notation.** In this paper, we use plain letters such as $x$ to denote scalars, lowercase bold letters such as $\mathbf{x}$ to denote vectors and uppercase bold letters such as $\mathbf{X}$ to denote matrices. For a vector $\mathbf{x}$, $\|\mathbf{x}\|_2$ denotes its $\ell_2$-norm. The weighted $\ell_2$-norm associated with a positive-definite matrix $\mathbf{A}$ is defined as $\|\mathbf{x}\|_{\mathbf{A}} = \sqrt{\mathbf{x}^\top \mathbf{A}\mathbf{x}}$. For two symmetric matrices $\mathbf{A}$ and $\mathbf{B}$, we use $\mathbf{A} \succeq \mathbf{B}$ to denote $\mathbf{A} - \mathbf{B}$ is positive semidefinite. We use $\mathbb{1}$ to denote the indicator function and $\mathbf{0}$ to denote the zero vector. For two actions $a$, $b$, we use $a \succ b$ to denote $a$ is more preferable to $b$. For a postive integer $N$, we use $[N]$ to denote $\{1, 2, \ldots, N\}$. We use standard asymptotic notations including $O(\cdot), \Omega(\cdot), \Theta(\cdot)$, and $\widetilde{O}(\cdot), \widetilde{\Omega}(\cdot), \widetilde{\Theta}(\cdot)$ will hide logarithmic factors.

## 2 RELATED WORK

**Bandits with Adversarial Reward.** The multi-armed bandit problem, involving an agent making sequential decisions among multiple arms, has been studied with both stochastic rewards (Lai et al., 1985; Lai, 1987; Auer, 2002; Auer et al., 2002a; Kalyanakrishnan et al., 2012; Lattimore & Szepesvári, 2020; Agrawal & Goyal, 2012), and adversarial rewards (Auer et al., 2002b; Bubeck et al., 2012). Moreover, a line of works focuses on designing algorithms that can achieve near-optimal regret bounds for both stochastic bandits and adversarial bandits simultaneously (Bubeck & Slivkins, 2012; Seldin & Slivkins, 2014; Auer & Chiang, 2016; Seldin & Lugosi, 2017; Zimmert

& Seldin, 2019; Lee et al., 2021), which is known as "the best of both worlds" guarantee. Distinct from fully stochastic and fully adversarial models, Lykouris et al. (2018) studied a setting, where only a portion of the rewards is subject to corruption. They proposed an algorithm with a regret dependent on the corruption level $C$, defined as the cumulative sum of the corruption magnitudes in each round. Their result is $C$ times worse than the regret without corruption. Gupta et al. (2019) improved the result by providing a regret guarantee comprising two terms, a corruption-independent term that matches the regret lower bound without corruption, and a corruption-dependent term that is linear in $C$. In addition, Gupta et al. (2019) proved a lower bound demonstrating the optimality of the linear dependency on $C$.

**Contextual Bandits with Corruption.** Li et al. (2019) studied stochastic linear bandits with corruption and presented an instance-dependent regret bound linearly dependent on the corruption level $C$. Bogunovic et al. (2021) studied the same problem and proposed an algorithm with near-optimal regret in the non-corrupted case. Lee et al. (2021) studied this problem in a different setting, where the adversarial corruptions are generated through the inner product of a corrupted vector and the context vector. For linear contextual bandits, Bogunovic et al. (2021) proved that under an additional context diversity assumption, the regret of a simple greedy algorithm is nearly optimal with an additive corruption term. Zhao et al. (2021) and Ding et al. (2022) extended the OFUL algorithm (Abbasi-Yadkori et al., 2011) and proved a regret with a corruption term polynomially dependent on the total number of rounds $T$. He et al. (2022) proposed an algorithm for known corruption level $C$ to remove the polynomial dependency on $T$ in the corruption term, which only has a linear dependency on $C$. They also proved a lower bound showing the optimality of linear dependency on $C$ for linear contextual bandits with a known corruption level. Additionally, He et al. (2022) extended the proposed algorithm to an unknown corruption level and provided a near-optimal performance guarantee that matches the lower bound. For more extensions, Kuroki et al. (2023) studied best-of-both-worlds algorithms for linear contextual bandits. Ye et al. (2023) proposed a corruption robust algorithm for nonlinear contextual bandits.

**Dueling Bandits and Logistic Bandits.** The dueling bandit model was first proposed in Yue et al. (2012). Compared with bandits, the agent will select two arms and receive the preference feedback between the two arms from the environment. For general preference, there may not exist the "best" arm that always wins in the pairwise comparison. Therefore, various alternative winners are considered, including Condorcet winner (Zoghi et al., 2014; Komiyama et al., 2015), Copeland winner (Zoghi et al., 2015; Wu & Liu, 2016; Komiyama et al., 2016), Borda winner (Jamieson et al., 2015; Falahatgar et al., 2017; Heckel et al., 2018; Saha et al., 2021; Wu et al., 2023) and von Neumann winner (Ramamohan et al., 2016; Dudík et al., 2015; Balsubramani et al., 2016), along with their corresponding performance metrics. To handle potentially large action space or context information, Saha (2021) studied a structured contextual dueling bandit setting. In this setting, each arm possesses an unknown intrinsic reward. The comparison is determined based on a logistic function of the relative rewards. In a similar setting, Bengs et al. (2022) studied contextual linear stochastic transitivity model with contextualized utilities. Di et al. (2023) proposed a layered algorithm with variance aware regret bound. Another line of works does not make the reward assumption. Instead, they assume the preference feedback can be represented by a function class. Saha & Krishnamurthy (2022) designed an algorithm that achieves the optimal regret for $K$-armed contextual dueling bandit problem. Sekhari et al. (2023) studied contextual dueling bandits in a more general setting and proposed an algorithm the provides guarantees for both regret and the number of queries. Another related area of research is the logistic bandits, where the agent selects one arm in each round and receives a Bernoulli reward. Faury et al. (2020) studied the dependency with respect to the degree of non-linearity of the logistic function $\kappa$. They proposed an algorithm with no dependency in $\kappa$. Abeille et al. (2021) further improved the dependency on $\kappa$ and proved a problem dependent lower bound. Faury et al. (2022) proposed a computationally efficient algorithm with regret performance still matching the lower-bound proved in Abeille et al. (2021).

**Dueling Bandits with Adversarial Feedback.** A line of work has focused on dueling bandits with adversarial feedback or corruption. Gajane et al. (2015) studied a fully adversarial utility-based version of dueling bandits, which was proposed in Ailon et al. (2014). Saha et al. (2021) considered the Borda regret for adversarial dueling bandits without the assumption of utility. In a setting parallel to that in Lykouris et al. (2018); Gupta et al. (2019), Agarwal et al. (2021) studied $K$-armed dueling bandits in a scenario where an adversary has the capability to corrupt part of the feedback received by the learner. They designed an algorithm whose regret comprises two terms: one that is optimal in uncorrupted scenarios, and another that is linearly dependent on the

total times of adversarial feedback $C$. Later on, Saha & Gaillard (2022) achieved "best-of-both world" result for noncontextual dueling bandits and improved the adversarial term of Agarwal et al. (2021) in the same setting. For contextual dueling bandits, Wu et al. (2023) proposed an EXP3-type algorithm for the adversarial linear setting using Borda regret. For a comparison of the most related works for robust bandits and dueling bandits, please refer to Table 1. In this paper, we study the influence of adversarial feedback within contextual dueling bandits, particularly in a setting where only a minority of the feedback is adversarial. Compared to previous studies, most studies have focused on the multi-armed dueling bandit framework without integrating context information. The notable exception is Wu et al. (2023); however, this study does not provide guarantees regarding the dependency on the number of adversarial feedback instances.

## 3 PRELIMINARIES

In this work, we study linear contextual dueling bandits with adversarial feedback. In each round $t \in [T]$, the agent observes the context information $x_t$ from a context set $\mathcal{X}$ and the corresponding action set $\mathcal{A}$. Utilizing this context information, the agent selects two actions, $a_t$ and $b_t$. Subsequently, the environment will generate a binary feedback (i.e., preference label) $l_t = \mathbb{1}(a_t \succ b_t) \in \{0, 1\}$ indicating the preferable action. We assume the existence of a reward function $r^*(x, a)$ dependent on the context information $x$ and action $a$, and a monotonically increasing link function $\sigma$ satisfying $\sigma(x) + \sigma(-x) = 1$. The preference probability will be determined by the link function and the difference between the rewards of the selected arms, i.e.,

$$\mathbb{P}(a \succ b|x) = \sigma\big(r^*(x, a) - r^*(x, b)\big). \tag{3.1}$$

We assume that the reward function is linear with respect to some known feature map $\phi(x, a)$. To be more specific, we make the following assumption:

**Assumption 3.1.** Let $\phi : \mathcal{X} \times \mathcal{A} \to \mathbb{R}^d$ be a known feature map, with $\|\phi(x, a)\|_2 \leq 1$ for any $(x, a) \in \mathcal{X} \times \mathcal{A}$. We define the reward function $r_{\boldsymbol{\theta}}$ parameterized by $\boldsymbol{\theta} \in \Theta$, with $r_{\boldsymbol{\theta}}(x, a) = \langle \boldsymbol{\theta}, \phi(x, a) \rangle$. Moreover, there exists $\boldsymbol{\theta}^*$ satisfying $r_{\boldsymbol{\theta}^*} = r^*$. For all with $\boldsymbol{\theta} \in \Theta$, $\|\boldsymbol{\theta}\|_2 \leq B$.

Similar linear assumptions have been made in the literature of dueling bandits (Saha, 2021; Bengs et al., 2022; Xiong et al., 2023). We also make an assumption on the derivative of the link function, which is common in the study of generalized linear models for bandits (Filippi et al., 2010).

**Assumption 3.2.** The link function $\sigma$ is differentiable. Furthermore, its first-order derivative satisfies that there exists a constant $\kappa > 0$ such that

$$\dot{\sigma}\big(\langle \phi(x, a) - \phi(x, b), \boldsymbol{\theta} \rangle\big) \geq \kappa,$$

for all $x \in \mathcal{X}, a, b \in \mathcal{A}, \boldsymbol{\theta} \in \Theta$.

In our setting, however, the agent does not directly observe the true binary feedback. Instead, an adversary will see both the choice of the agent and the true feedback. Based on the information, the adversary can decide whether to corrupt the binary feedback or not.[1] We represent the adversary's decision in round $t$ by an adversarial indicator $c_t$, which takes values from the set $\{0, 1\}$. If the adversary chooses not to corrupt the result, we have $c_t = 0$. Otherwise, we have $c_t = 1$, which means adversarial feedback in this round. As a result, the agent will observe a flipped preference label, i.e., the observation $o_t = 1 - l_t$. We define $C$ as the total level of adversarial feedback, i.e.,

$$\sum_{t=1}^{T} c_t \leq C.$$

**Remark 3.3.** There are two commonly used corruption models for bandits. One is the total budget model (Lykouris et al., 2018), where in each round $t$, the agent selects an action $a_t$ and the environment generates a numerical reward $r_t(a_t)$. The adversary observes the reward and returns a corrupted reward $\bar{r}_t$. The corruption level $C$ is defined by $\sum_{t=1}^{T} |r_t(a_t) - \bar{r}_t| \leq C$. Another considers the number of corrupted rounds (Zhang et al., 2021). In our setting, we consider the label-flipping attack. Thus, the magnitude of adversarial feedback is always 1 and these two types of corruption models are equivalent. Moreover, adversarial feedback in our setting involves comparing two arms, whereas in bandits it pertains to the reward of a single arm. The only previous work that studied label-flipping is (Agarwal et al., 2021), where the adversary cannot observe the action selected by the agent. In contrast, our setting focuses on scenarios where this information is available to adversaries, which is common in many real-life applications. We use the term "adversarial feedback" to differentiate our work from prior studies on corrupted or adversarial reward settings.

---

[1]Such adversary is referred to as strong adversary (He et al., 2022), compared with the weak adversary who cannot obtain the information before the decision.

As the context is changing, the optimal action is different in each round, denoted by $a_t^* = \arg\max_{a \in \mathcal{A}} r^*(x_t, a)$. The goal of our algorithm is to minimize the cumulative gap between the rewards of both selected actions and the optimal action

$$\text{Regret}(T) = \sum_{t=1}^{T} 2r^*(x_t, a_t^*) - r^*(x_t, a_t) - r^*(x_t, b_t). \tag{3.2}$$

This regret definition is the same as that in Saha (2021) and the average regret defined in Bengs et al. (2022). It is typically stronger than weak regret defined in Bengs et al. (2022), which only considers the reward gap of the better action.

## 4 ALGORITHM

In this section, we present our new algorithm RCDB, designed for learning contextual linear dueling bandits. The main algorithm is illustrated in Algorithm 1. At a high level, we incorporate uncertainty-dependent weighting into the Maximum Likelihood Estimator (MLE) to counter adversarial feedback. Specifically, in each round $t \in [T]$, we construct the estimator of parameter $\boldsymbol{\theta}$ by solving the following equation:

$$\lambda\kappa\boldsymbol{\theta} + \sum_{i=1}^{t-1} w_i \big(\sigma(\boldsymbol{\phi}_i^\top \boldsymbol{\theta}) - o_i\big)\boldsymbol{\phi}_i = \mathbf{0}, \tag{4.1}$$

where we denote $\boldsymbol{\phi}_i = \boldsymbol{\phi}(x_i, a_i) - \boldsymbol{\phi}(x_i, b_i)$ for simplicity, $w_i$ is the uncertainty weight we are going to choose. To obtain an intuitive understanding of our weight, we consider any action-observation sequence $(x_1, a_1, b_1, o_1, x_2, a_2, b_2, o_2, \ldots, x_t, a_t, b_t, o_t)$ up to round $t$. For simplicity, we denote $\mathcal{F}_t = \sigma(x_1, a_1, b_1, o_1, x_2, a_2, b_2, o_2, \ldots, x_t, a_t, b_t)$ as the filtration. Suppose the estimated parameter $\boldsymbol{\theta}_t$ is the solution to the unweighted version equation of (4.1), i.e.,

$$\lambda\kappa\boldsymbol{\theta}_t + \sum_{i=1}^{t} \big(\sigma(\boldsymbol{\phi}_i^\top \boldsymbol{\theta}_t) - o_i\big)\boldsymbol{\phi}_i = \mathbf{0}. \tag{4.2}$$

When we receive $\boldsymbol{\phi}_t = \boldsymbol{\phi}(x_t, a_t) - \boldsymbol{\phi}(x_t, b_t)$, the probability of receiving $l_t = 1$ can be estimated by $\sigma(\boldsymbol{\phi}_t^\top \boldsymbol{\theta}_t)$. We consider the conditional variance of the estimated probability $\sigma(\boldsymbol{\phi}_t^\top \boldsymbol{\theta}_t)$ in round $t$, i.e., $\text{Var}\big[\sigma(\boldsymbol{\phi}_t^\top \boldsymbol{\theta}_t)|\mathcal{F}_t\big]$, involving a posterior estimate of the prediction's variance. Intuitively, even without the weighting, we can show that the solution of (4.2), i.e., $\boldsymbol{\theta}_t$, will approach $\boldsymbol{\theta}^*$, using the arguments similar to Lemma 5.1, what we will present next. This inspires us to consider the approximation of Taylor's expansion:

$$
\begin{aligned}
\mathbb{E}\big[\sigma(\boldsymbol{\phi}_t^\top \boldsymbol{\theta}_t)|\mathcal{F}_t\big] &\approx \mathbb{E}\big[\sigma(\boldsymbol{\phi}_t^\top \boldsymbol{\theta}^*) + \sigma'(\boldsymbol{\phi}_t^\top \boldsymbol{\theta}^*)\boldsymbol{\phi}_t^\top (\boldsymbol{\theta}_t - \boldsymbol{\theta}^*)|\mathcal{F}_t\big] \\
&= \mathbb{E}\big[\underbrace{\sigma(\boldsymbol{\phi}_t^\top \boldsymbol{\theta}^*) - \sigma'(\boldsymbol{\phi}_t^\top \boldsymbol{\theta}^*)\boldsymbol{\phi}_t^\top \boldsymbol{\theta}^*}_{\mathcal{F}_t-\text{measurable}} |\mathcal{F}_t\big] + \mathbb{E}\big[\sigma'(\boldsymbol{\phi}_t^\top \boldsymbol{\theta}^*)\boldsymbol{\phi}_t^\top \boldsymbol{\theta}_t|\mathcal{F}_t\big].
\end{aligned}
$$

Moreover, using the Taylor's expansion to (4.2), we have

$$
\begin{aligned}
\mathbf{0} &= \lambda\kappa\boldsymbol{\theta}_t + \sum_{i=1}^{t} \big(\sigma(\boldsymbol{\phi}_i^\top \boldsymbol{\theta}_t) - o_i\big)\boldsymbol{\phi}_i \\
&\approx \Big(\lambda\kappa\mathbf{I} + \sum_{i=1}^{t}\sigma'(\boldsymbol{\phi}_i^\top \boldsymbol{\theta}^*)\boldsymbol{\phi}_i\boldsymbol{\phi}_i^\top\Big)\boldsymbol{\theta}_t + \sum_{i=1}^{t} \big(\sigma(\boldsymbol{\phi}_i^\top \boldsymbol{\theta}^*) - o_i\big)\boldsymbol{\phi}_i - \sum_{i=1}^{t}\sigma'(\boldsymbol{\phi}_i^\top \boldsymbol{\theta}^*)\boldsymbol{\phi}_i\boldsymbol{\phi}_i^\top \boldsymbol{\theta}^*.
\end{aligned}
$$

Let $\boldsymbol{\Lambda}_t = \lambda\kappa\mathbf{I} + \sum_{i=1}^{t}\sigma'(\boldsymbol{\phi}_i^\top \boldsymbol{\theta}^*)\boldsymbol{\phi}_i\boldsymbol{\phi}_i^\top$, we have

$$
\begin{aligned}
\boldsymbol{\theta}_t &\approx \boldsymbol{\Lambda}_t^{-1}\Big[\sum_{i=1}^{t}\sigma'(\boldsymbol{\phi}_i^\top \boldsymbol{\theta}^*)\boldsymbol{\phi}_i\boldsymbol{\phi}_i^\top \boldsymbol{\theta}^* - \sum_{i=1}^{t} \big(\sigma(\boldsymbol{\phi}_i^\top \boldsymbol{\theta}^*) - o_i\big)\boldsymbol{\phi}_i\Big] \\
&= \underbrace{\boldsymbol{\Lambda}_t^{-1}\Big[\sum_{i=1}^{t}\sigma'(\boldsymbol{\phi}_i^\top \boldsymbol{\theta}^*)\boldsymbol{\phi}_i\boldsymbol{\phi}_i^\top \boldsymbol{\theta}^* - \sum_{i=1}^{t-1} \big(\sigma(\boldsymbol{\phi}_i^\top \boldsymbol{\theta}^*) - o_i\big)\boldsymbol{\phi}_i - \sigma(\boldsymbol{\phi}_t^\top \boldsymbol{\theta}^*)\Big]}_{\mathcal{F}_t-\text{measurable}} + o_t\boldsymbol{\Lambda}_t^{-1}\boldsymbol{\phi}_t
\end{aligned}
$$

Therefore, applying the pulling-out-known-factor property of the conditional expectation, the $\mathcal{F}_t$-measurable part will cancel out when calculating the conditional variance. Then, we can approximate the variance of the estimated preference probability by

$$
\begin{aligned}
\text{Var}\big[\sigma(\boldsymbol{\phi}_t^\top \boldsymbol{\theta}_t)|\mathcal{F}_t\big] &= \mathbb{E}\big[\big(\sigma(\boldsymbol{\phi}_t^\top \boldsymbol{\theta}_t) - \mathbb{E}\big[\sigma(\boldsymbol{\phi}_t^\top \boldsymbol{\theta}_t)|\mathcal{F}_t\big]\big)^2|\mathcal{F}_t\big] \\
&\approx \mathbb{E}\Big[\big(\mathbb{E}\big[o_t\sigma'(\boldsymbol{\phi}_t^\top \boldsymbol{\theta}^*)\boldsymbol{\phi}_t^\top \boldsymbol{\Lambda}_t^{-1}\boldsymbol{\phi}_t|\mathcal{F}_t\big]\big)^2\Big|\mathcal{F}_t\Big] \\
&\leq \mathbb{E}\big[o_t[\sigma'(\boldsymbol{\phi}_t^\top \boldsymbol{\theta}^*)]^2\|\boldsymbol{\phi}_t\|_{\boldsymbol{\Lambda}_t^{-1}}^2|\mathcal{F}_t\big] \leq [\sigma'(\boldsymbol{\phi}_t^\top \boldsymbol{\theta}^*)]^2\|\boldsymbol{\phi}_t\|_{\boldsymbol{\Lambda}_t^{-1}}^2,
\end{aligned}
$$

---

**Algorithm 1** Robust Contextual Dueling Bandit (RCDB)

---

1: **Require:** $\alpha > 0$, Regularization parameter $\lambda$, confidence radius $\beta$.
2: **for** $t = 1, \ldots, T$ **do**
3:    Compute $\boldsymbol{\Sigma}_t = \lambda \mathbf{I} + \sum_{i=1}^{t-1} w_i \big( \boldsymbol{\phi}(x_i, a_i) - \boldsymbol{\phi}(x_i, b_i) \big) \big( \boldsymbol{\phi}(x_i, a_i) - \boldsymbol{\phi}(x_i, b_i) \big)^\top$.
4:    Calculate the MLE $\boldsymbol{\theta}_t$ by solving the following equation:

$$\lambda \kappa \boldsymbol{\theta} + \sum_{i=1}^{t-1} w_i \Big[ \sigma \Big( \big( \boldsymbol{\phi}(x_i, a_i) - \boldsymbol{\phi}(x_i, b_i) \big)^\top \boldsymbol{\theta} \Big) - o_i \Big] \big( \boldsymbol{\phi}(x_i, a_i) - \boldsymbol{\phi}(x_i, b_i) \big) = \mathbf{0}. \quad (4.4)$$

5:    Observe the context vector $x_t$.
6:    Choose $a_t, b_t = \operatorname{argmax}_{a,b} \Big\{ \big( \boldsymbol{\phi}(x_t, a) + \boldsymbol{\phi}(x_t, b) \big)^\top \boldsymbol{\theta}_t + \beta \big\| \boldsymbol{\phi}(x_t, a) - \boldsymbol{\phi}(x_t, b) \big\|_{\boldsymbol{\Sigma}_t^{-1}} \Big\}$.
7:    The adversary sees the feedback $l_t = \mathbb{1}(a_t \succ b_t)$ and decides the indicator $c_t$. Observe $o_t = l_t$ when $c_t = 0$, otherwise observe $o_t = 1 - l_t$.
8:    Set weight $w_t$ as (4.3).
9: **end for**

---

where the first inequality holds due to the Jensen's inequality and $o_t^2 = o_t$, and the last inequality holds due to $\mathbb{E}[o_t | \mathcal{F}_t] \leq 1$. Using $\sigma'(\boldsymbol{\phi}_t^\top \boldsymbol{\theta}^*) \leq 1$, $\boldsymbol{\Lambda}_t \geq \kappa \boldsymbol{\Sigma}_{t+1} \geq \kappa \boldsymbol{\Sigma}_t$, where $\boldsymbol{\Sigma}_t$ is defined in Line 3 of Algorithm 1, we can see that $\operatorname{Var}\big[ \sigma(\boldsymbol{\phi}_t^\top \boldsymbol{\theta}_t) | \mathcal{F}_t \big] \leq \kappa^{-1} \| \boldsymbol{\phi}_t \|_{\boldsymbol{\Sigma}_t^{-1}}^2$. Since higher variance leads to larger uncertainty, which harms the credibility of the data, it is natural to assign a smaller weight to the data with high uncertainty. Thus, we choose the weight to cancel out the uncertainty as follows

$$w_i = \min\{1, \alpha / \| \boldsymbol{\phi}_i \|_{\boldsymbol{\Sigma}_i^{-1}}\}, \quad (4.3)$$

where $\alpha / \| \boldsymbol{\phi}_i \|_{\boldsymbol{\Sigma}_i^{-1}}$ normalizes the variance of the estimated probability. To prevent excessively large weights, we apply truncation to this value. A similar weight has been used in He et al. (2022) for linear contextual bandits under corruption. Different from their setting where the weight is an estimate of the variance of the linear model, our weight is an estimate of a generalized linear model. Furthermore, by selecting a proper threshold parameter, e.g., $\alpha = \sqrt{d}/C$, the weighted MLE shares the same confidence radius with that of the no-adversary scenario.

**Remark 4.1.** Here, we use approximations to illustrate the motivation of our uncertainty-based weight. Rigorous proof for the algorithm's performance is presented in Section B.1, which relies solely on our specific choice of weights and does not use the approximation.

After constructing the estimator $\boldsymbol{\theta}_t$ from the weighted MLE, the sum of the estimated reward for each duel $(a, b)$ can be calculated as $\big( \boldsymbol{\phi}(x_t, a) + \boldsymbol{\phi}(x_t, b) \big)^\top \boldsymbol{\theta}_t$. To encourage the exploration of duel $(a, b)$ with high uncertainty during the learning process, we introduce an exploration bonus with the following $\beta \big\| \boldsymbol{\phi}(x_t, a) - \boldsymbol{\phi}(x_t, b) \big\|_{\boldsymbol{\Sigma}_t^{-1}}$, which follows a similar spirit to the bonus term in the context of linear bandit problems (Abbasi-Yadkori et al., 2011). However, the reward term and the bonus term exhibit different combinations of the feature maps $\boldsymbol{\phi}(x_t, a)$ and $\boldsymbol{\phi}(x_t, b)$, which is the key difference between bandits and dueling bandits. The selection of action pairs $(a, b)$ is subsequently determined by maximizing the estimated reward with the exploration bonus term, i.e.,

$$\big( \boldsymbol{\phi}(x_t, a) + \boldsymbol{\phi}(x_t, b) \big)^\top \boldsymbol{\theta}_t + \beta \big\| \boldsymbol{\phi}(x_t, a) - \boldsymbol{\phi}(x_t, b) \big\|_{\boldsymbol{\Sigma}_t^{-1}}.$$

More discussion about the selection rule was discussed in Appendix A of Di et al. (2023).

**Computational Complexity.** We assume there is a computation oracle to solve the optimization problems of the action selection over $\mathcal{A}$. A similar oracle is implicitly assumed in almost all existing works for solving standard linear bandit problems with infinite arms (e.g., (Abbasi-Yadkori et al., 2011; He et al., 2022)). In the special case where the decision set is finite, we can iterate across all actions, resulting in $O(k^2 d^2)$ complexity for each iteration, where $k$ is the number of actions, and $d$ is the feature dimension.

## 5 MAIN RESULTS

### 5.1 KNOWN NUMBER OF ADVERSARIAL FEEDBACK

At the center of our algorithm design is the uncertainty-weighted MLE. When faced with adversarial feedback, the estimation error of the weighted MLE $\boldsymbol{\theta}_t$ can be characterized by the following lemma.

**Lemma 5.1.** If we set $\beta = \sqrt{\lambda}B + \big(\alpha C + \sqrt{d \log((1 + 2T/\lambda)/\delta)}\big)/\kappa$, then with probability at least $1 - \delta$, for any $t \in [T]$, we have

$$\left\| \boldsymbol{\theta}_t - \boldsymbol{\theta}^* \right\|_{\boldsymbol{\Sigma}_t} \leq \beta.$$

The proof of this lemma is postponed to Section C.1.

**Remark 5.2.** If we set $\alpha = (\sqrt{d} + \sqrt{\lambda}B)/C$, then the bonus radius $\beta$ has no direct dependency on the number of adversarial feedback $C$. This observation plays a key role in proving the adversarial term in the regret without polynomial dependence on the total number of rounds $T$.

With Lemma 5.1, we can present the following regret guarantee of our algorithm RCDB in the dueling bandit framework.

**Theorem 5.3.** Under Assumption 3.1 and 3.2, let $0 < \delta < 1$, the total number of adversarial feedback be $C$. If we set the bonus radius to be

$$\beta = \sqrt{\lambda}B + \big(\alpha C + \sqrt{d \log((1 + 2T/\lambda)/\delta)}\big)/\kappa,$$

then with probability at least $1 - \delta$, the regret in the first $t$ rounds can be upper bounded by

$$\begin{aligned}
\text{Regret}(T) \leq\ & 4\big(\sqrt{\lambda}B + \alpha C/\kappa\big)\sqrt{dT \log(1 + 2T/\lambda)} \\
& + 4d\big(\sqrt{T}/\kappa + \sqrt{\lambda}B/\alpha + 4C/\kappa\big) \log\big((1 + 2T/\lambda)/\delta\big) \\
& + 4d^{1.5}\sqrt{\log^3\big((1 + 2T/\lambda)/\delta\big)}/(\alpha\kappa).
\end{aligned}$$

Moreover, if we set $\alpha = (\sqrt{d} + \sqrt{\lambda}B)/C$, $\lambda = 1/B^2$, the regret upper bound can be simplified to

$$\text{Regret}(T) = \widetilde{O}\big(d\sqrt{T}/\kappa + dC/\kappa\big).$$

**Remark 5.4.** The proof of Theorem 5.3 is postponed to Section B.1. Our regret bound consists of two terms. The first one is a $C$-independent term $\widetilde{O}(d\sqrt{T})$, which matches the lower bound $\widetilde{\Omega}(d\sqrt{T})$ proved in Bengs et al. (2022). This indicates that our result is optimal in scenarios without adversarial feedback ($C = 0$). Additionally, our result includes an additive term that is linearly dependent on the number of adversarial feedback $C$. When $C = O(\sqrt{T})$, the order of regret will be the same as the stochastic setting. It indicates the robustness of our algorithm to adversarial feedback. Additionally, the following theorem we present establishes a lower bound for this adversarial term, indicating that our dependency on the number of adversarial feedback $C$ and the context dimension $d$ is also optimal.

**Theorem 5.5.** For any dimension $d$, there exists an instance of dueling bandits with $|\mathcal{A}| = d$, such that any algorithm with the knowledge of the number of adversarial feedback $C$ must incur $\Omega(dC)$ regret with probability at least $1/2$.

**Remark 5.6.** The proof of Theorem 5.5 follows Bogunovic et al. (2021). In the constructed instances, only one action has reward 1, while others have 0. Compared with linear bandits, where the feedback is an exact reward, dueling bandits deal with the comparison between a pair of actions. A critical observation from our preference model, as formulated in (3.1), is that two actions with identical rewards result in a pair that is challenging to differentiate. The lower bound can be proved by corrupting every comparison into a random guess until the total times of adversarial feedback have been used up. For detailed proof, please refer to Section B.2. Our proved lower bound $\Omega(dC)$ shows that our result is nearly optimal because of the linear dependency on $C, d$ and only logarithmic dependency on the total number of rounds $T$.

## 5.2 Unknown Number of Adversarial Feedback

In our previous analysis, the selection of parameters depends on having prior knowledge of the total number of adversarial feedback $C$. In this subsection, we extend our previous result to address the challenge posed by an unknown number of adversarial feedback $C$. Our approach to tackle this uncertainty follows He et al. (2022), we introduce an adversarial tolerance threshold $\bar{C}$ for the adversary count. This threshold can be regarded as an optimistic estimator of the actual number of adversarial feedback $C$. Under this situation, the subsequent theorem provides an upper bound for regret of Algorithm 1 in the case of an unknown number of adversarial feedback $C$.

**Theorem 5.7.** Under Assumptions 3.1 and 3.2, if we set the the confidence radius as

$$\beta = \sqrt{\lambda}B + \left[\alpha\bar{C} + \sqrt{d\log\left((1+2T/\lambda)/\delta\right)}\right]/\kappa,$$

with the pre-defined adversarial tolerance threshold $\bar{C}$ and $\alpha = (\sqrt{d} + \sqrt{\lambda}B)/\bar{C}$, then with probability at least $1 - \delta$, the regret of Algorithm 1 can be upper bounded as following:

- If the actual number of adversarial feedback $C$ is smaller than the adversarial tolerance threshold $\bar{C}$, then we have

$$\text{Regret}(T) = \widetilde{O}\left(d\sqrt{T}/\kappa + d\bar{C}/\kappa\right).$$

- If the actual number of adversarial feedback $C$ is larger than the adversarial tolerance threshold $\bar{C}$, then we have $\text{Regret}(T) = O(T)$.

**Remark 5.8.** The COBE framework (Wei et al., 2022) converts any algorithm with the known adversarial level to an algorithm in the unknown case. However, such a framework only works for weak adversaries and does not work in our strong adversary setting. In fact, He et al. (2022) proved that any algorithm cannot simultaneously achieve near-optimal regret when uncorrupted and maintain sublinear regret with corruption level $C = \Omega(\sqrt{T})$. Therefore, there exists a trade-off between robust adversarial defense and near-optimal algorithmic performance, which is very common in dealing with strong adversaries (He et al., 2022; Ye et al., 2023). Our algorithm achieves the same nearly optimal $\widetilde{O}(d\sqrt{T})$ regret as the no-adversary case even when $C = \Theta(\sqrt{T})$, which indicates that our results are optimal in the presence of an unknown number of adversarial feedback.

# 6 EXPERIMENTS

## 6.1 EXPERIMENT SETUP

**Preference Model.** We study the effect of adversarial feedback with the preference model determined by (3.1), where $\sigma(x) = 1/(1 + e^{-x})$. We randomly generate the underlying parameter in $[-0.5, 0.5]^d$ and normalize it to be a vector with $\|\boldsymbol{\theta}^*\|_2 = 2$. Then, we set it to be the underlying parameter and construct the reward utilized in the preference model as $r^*(x, a) = \langle \boldsymbol{\theta}^*, \boldsymbol{\phi}(x, a) \rangle$. We set the action set $\mathcal{A} = \left\{-1/\sqrt{d}, 1/\sqrt{d}\right\}^d$. For simplicity, we assume $\boldsymbol{\phi}(x, a) = a$. In our experiment, we set the dimension $d = 5$, with the size of action set $|\mathcal{A}| = 2^d = 32$.

**Adversarial Attack Methods.** We study the performance of our algorithm using different adversarial attack methods. We categorize the first two methods as "weak" primarily because the adversary in these scenarios does not utilize information about the agent's actions. In contrast, we classify the latter two methods as "strong" attacks. In these cases, the adversary leverages a broader scope of information, including knowledge of the actions selected by the agent and the true preference model. This enables it to devise more targeted adversarial methods.

- "Greedy Attack": The adversary will flip the preference label for the first $C$ rounds. After that, it will not corrupt the result anymore.
- "Random Attack": In each round, the adversary will flip the preference label with the probability of $0 < p < 1$, until the times of adversarial feedback reach $C$.
- "Adversarial Attack": The adversary can have access to the true preference model. It will only flip the preference label when it aligns with the preference model, i.e., the probability for the preference model to make that decision is larger than 0.5, until the times of adversarial feedback reach $C$.
- "Misleading Attack": The adversary selects a suboptimal action. It will make sure this arm is always the winner in the comparison until the times of adversarial feedback reach $C$. In this way, it will mislead the agent to believe this action is the optimal one.

**Experiment Setup.** For each experiment instance, we simulate the interaction with the environment for $T = 2000$ rounds. In each round, the feedback for the action pair selected by the algorithm is generated according to the defined preference model. Subsequently, the adversary observes both the selected actions and their corresponding feedback and then engages in one of the previously described adversarial attack methods. We report the regret defined in (3.2) averaged across 10 random runs.

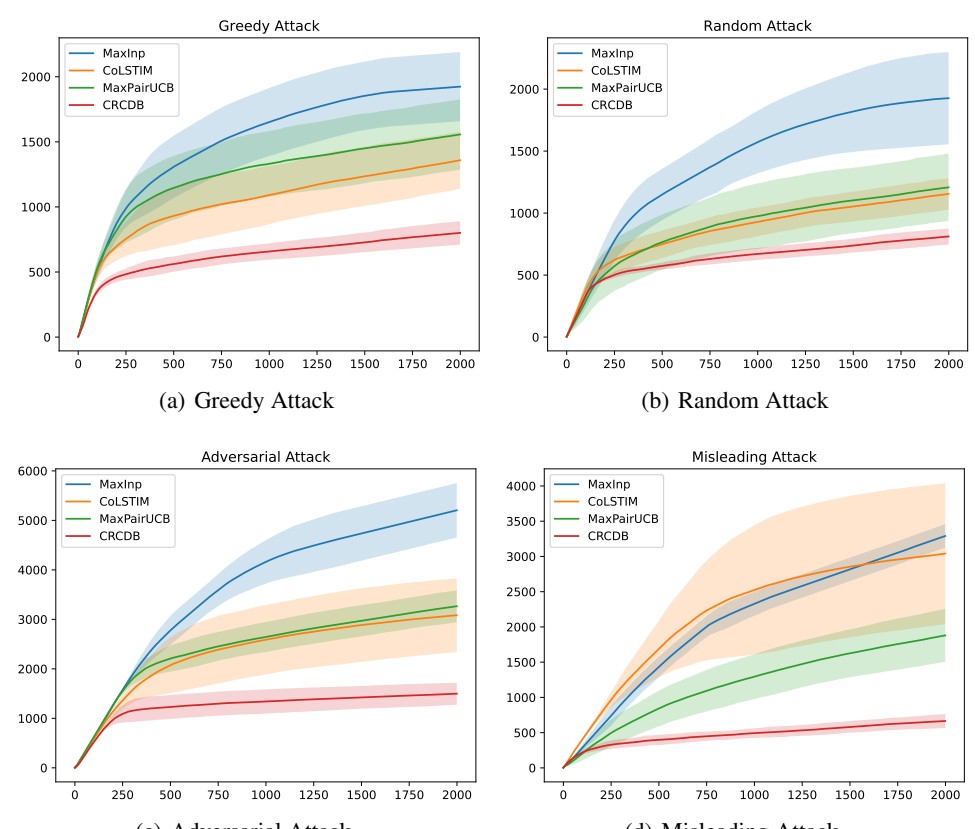

Figure 1: Comparison of `RCDB` (Our Algorithm 1), `MaxInp` (Saha, 2021), `CoLSTIM` (Bengs et al., 2022) and `MaxPairUCB` (Di et al., 2023). We report the cumulative regret with various adversarial attack methods (Greedy, Random, Adversarial, Misleading). For the baselines, the parameters are carefully tuned to achieve better results with different attack methods. The total number of adversarial feedback is $C = \lceil \sqrt{T} \rceil$.

## 6.2 PERFORMANCE COMPARISON

We first introduce the algorithms studied in this section.

- `MaxInP`: Maximum Informative Pair by Saha (2021). It involves maintaining a standard MLE. With the estimated model, it then identifies a set of promising arms possible to beat the rest. The selection of arm pairs is then strategically designed to maximize the uncertainty in the difference between the two arms within this promising set, referred to as "maximum informative".
- `CoLSTIM`: The method by Bengs et al. (2022). It involves maintaining a standard MLE for the estimated model. Based on this model, the first arm is selected as the one with the highest estimated reward, implying it is the most likely to prevail over competitors. The second arm is selected to be the first arm's toughest competitor, with an added uncertainty bonus.
- `MaxPairUCB`: This algorithm was proposed in Di et al. (2023). It uses the regularized MLE to estimate the parameter $\theta^*$. Then it selects the actions based on a symmetric action selection rule, i.e. the actions with the largest estimated reward plus some uncertainty bonus.
- `RCDB`: Algorithm 1 proposed in this paper. The key difference from the other algorithms is the use of uncertainty weight in the calculation of MLE (4.4). The we use the same symmetric action selection rule as `MaxPairUCB`. Our experiment results show that the uncertainty weight is critical in the face of adversarial feedback.

Our results are demonstrated in Figure 1. In Figure 1(a) and Figure 1(b), we observe scenarios where the adversary is "weak" due to the lack of access to information regarding the selected actions and the underlying preference model. Notably, in these situations, our algorithm `RCDB` outperforms all other baseline algorithms, demonstrating its robustness. Among the other algorithms, `CoLSTIM` performs as the strongest competitor.

In Figure 1(c), the adversary employs a 'stronger' adversarial method. Due to the inherent randomness of the model, some labels may naturally be 'incorrect'. An adversary with knowledge of the selected actions and the preference model can strategically neglect these naturally incorrect labels and selectively flip the others. This method proves catastrophic for algorithms to learn the true model, as it results in the agent encountering only incorrect preference labels at the beginning. Our results indicate that this leads to significantly higher regret. However, it's noteworthy that our algorithm `RCDB` demonstrates considerable robustness.

In Figure 1(d), the adversary employs a strategy aimed at misleading algorithms into believing a suboptimal action is the best choice. The algorithm `CoLSTIM` appears to be the most susceptible to being cheated by this method. Despite the deployment of 'strong' adversarial methods, as shown in both Figure 1(c) and Figure 1(d), our algorithm, `RCDB`, consistently demonstrates exceptional robustness against these attacks. A significant advantage of `RCDB` lies in that our parameter is selected solely based on the number of adversarial feedback $C$, irrespective of the nature of the adversarial methods employed. This contrasts with other algorithms where parameter tuning must be specifically adapted for each distinct adversarial method.

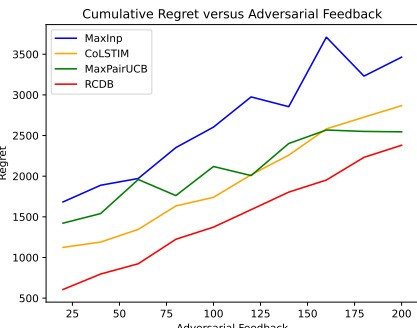

Figure 2: The relationship between cumulative regret and the number of adversarial feedback $C$. For this specific experiment, we employ the "greedy attack" method to generate the adversarial feedback. $C$ is selected from the set $[20, 40, 60, 80, 100, 120, 140, 160, 180, 200]$ (10 adversarial levels).

### 6.3 ROBUSTNESS TO DIFFERENT NUMBERS OF ADVERSARIAL FEEDBACK

In this section, we test the performance of algorithms with increasing times of adversarial feedback. Our results show a linear dependency on the number of adversarial feedback $C$, which is consistent with the theoretical results we have proved in Theorem 5.3 and 5.5. In comparison to other algorithms, `RCDB` demonstrates superior robustness against adversarial feedback, as evidenced by its notably smaller regret.

## 7 CONCLUSION

In this paper, we focus on the contextual dueling bandit problem from adversarial feedback. We introduce a novel algorithm, `RCDB`, which utilizes an uncertainty-weighted Maximum Likelihood Estimator (MLE) approach. This algorithm not only achieves optimal theoretical results in scenarios with and without adversarial feedback but also demonstrates superior performance with synthetic data. For future direction, we aim to extend our uncertainty-weighted method to encompass more general settings involving preference-based data. A particularly promising future direction of our research lies in addressing adversarial feedback within the process of aligning large language models using Reinforcement Learning from Human Feedback (RLHF).

**Limitations and Future Works.** We assume that the reward is linear with respect to some known feature maps. Although this setting is common in the literature, we observe that some recent works on dueling bandits can deal with nonlinear rewards (Li et al., 2024; Verma et al., 2024). Recently, Verma et al. (2024) studied the problem of approximating reward models using neural networks, addressing nonlinear rewards for dueling bandits. It is an interesting future direction to design robust algorithms for nonlinear reward functions, such as with neural networks. Another assumption concerns the lower bound of the derivative of the link function. Notably, in the logistic bandit model, which shares similarities with our setting through Bernoulli variables, some work (Abeille et al., 2021; Faury et al., 2022) can improve the dependency of $\kappa$ from $1/\kappa$ to $\sqrt{\kappa}$. A similar improvement might be achieved in our setting as well.

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

BROADER IMPACT

This paper studies contextual dueling bandits with adversarial feedback. Our primary objective is to propel advancements in bandit theory by introducing a more robust algorithm backed by solid theoretical guarantees. The uncertainty-weighted approach we have developed for dueling bandits holds significant potential to address the issue of adversarial feedback in preference-based data, which could be instrumental in enhancing the robustness of generative models against adversarial attacks, thereby contributing positively to the societal impact and reliability of machine learning applications.

# A  ROADMAP OF THE PROOF

## A.1  UNCERTAINTY-WEIGHTED MLE WITH ADVERSARIAL FEEDBACK

In this section, we offer an overview of the proof for Lemma 5.1. The general proof idea for the uncertainty-weighted MLE with adversarial feedback lies in decomposing the estimation error into three terms, a stochastic error term, an adversarial term, and an additional regularization term. Following the analysis of standard (weighted) MLE (Li et al., 2017), we introduce an auxiliary function:

$$G_t(\boldsymbol{\theta}) = \lambda\kappa\boldsymbol{\theta} + \sum_{i=1}^{t-1} w_i \Big[ \sigma\Big( \big(\boldsymbol{\phi}(x_i, a_i) - \boldsymbol{\phi}(x_i, b_i)\big)^\top \boldsymbol{\theta} \Big)$$
$$- \sigma\Big( \big(\boldsymbol{\phi}(x_i, a_i) - \boldsymbol{\phi}(x_i, b_i)\big)^\top \boldsymbol{\theta}^* \Big) \Big] \big(\boldsymbol{\phi}(x_i, a_i) - \boldsymbol{\phi}(x_i, b_i)\big).$$

It satisfies two conditions: First, for the true parameter value $\boldsymbol{\theta}^*$, $G_t(\boldsymbol{\theta}^*)$ has a simple expression, i.e.,

$$G_t(\boldsymbol{\theta}^*) = \lambda\kappa\boldsymbol{\theta}^*.$$

Second, according to (4.4), we can get the value of function $G_t$ at the MLE $\boldsymbol{\theta}_t$,

$$G_t(\boldsymbol{\theta}_t) = \sum_{i=1}^{t-1} w_i \gamma_i \big(\boldsymbol{\phi}(x_i, a_i) - \boldsymbol{\phi}(x_i, b_i)\big), \tag{A.1}$$

where $\gamma_i = o_i - \sigma\big((\boldsymbol{\phi}(x_i, a_i) - \boldsymbol{\phi}(x_i, b_i))^\top \boldsymbol{\theta}^*\big)$. To connect the desired estimation error with the function $G_t$, we use the mean value theorem. This leads to an upper bound of the estimation error:

$$\|\boldsymbol{\theta}_t - \boldsymbol{\theta}^*\|_{\boldsymbol{\Sigma}_t} \leq \frac{1}{\kappa} \big\| G_t(\boldsymbol{\theta}_t) - G_t(\boldsymbol{\theta}^*) \big\|_{\boldsymbol{\Sigma}_t^{-1}}$$
$$\leq \underbrace{\frac{1}{\kappa}\lambda \|\boldsymbol{\theta}^*\|_{\boldsymbol{\Sigma}_t^{-1}}}_{\text{Regularization term}} + \underbrace{\frac{1}{\kappa}\big\| G_t(\boldsymbol{\theta}_t) \big\|_{\boldsymbol{\Sigma}_t^{-1}}}_{I_1}.$$

For term $I_1$, we can decompose the summation in (A.1) based on the adversarial feedback $c_t$, i.e.,

$$G_t(\boldsymbol{\theta}_t) = \sum_{i < t : c_i = 0} w_i \gamma_i \big(\boldsymbol{\phi}(x_i, a_i) - \boldsymbol{\phi}(x_i, b_i)\big) + \underbrace{\sum_{i < t : c_i = 1} w_i \gamma_i \big(\boldsymbol{\phi}(x_i, a_i) - \boldsymbol{\phi}(x_i, b_i)\big)}_{I_2},$$

where $I_2$ can be further decomposed as

$$I_2 = \sum_{i < t : c_i = 1} w_i \epsilon_i \big(\boldsymbol{\phi}(x_i, a_i) - \boldsymbol{\phi}(x_i, b_i)\big) + \sum_{i < t : c_i = 1} w_i (\gamma_i - \epsilon_i) \big(\boldsymbol{\phi}(x_i, a_i) - \boldsymbol{\phi}(x_i, b_i)\big).$$

where $\epsilon_i = l_i - \sigma\big((\boldsymbol{\phi}(x_i, a_i) - \boldsymbol{\phi}(x_i, b_i))^\top \boldsymbol{\theta}^*\big)$. With our notation of adversarial feedback, when $c_i = 0$, we have $\gamma_i = \epsilon_i$. Therefore, we have $|\gamma_i - \epsilon_i| \leq 1$ and

$$I_1 \leq \underbrace{\frac{1}{\kappa}\Big\| \sum_{i=1}^{t-1} w_i \epsilon_i \big(\boldsymbol{\phi}(x_i, a_i) - \boldsymbol{\phi}(x_i, b_i)\big) \Big\|_{\boldsymbol{\Sigma}_t^{-1}}}_{\text{Stochastic term}} + \underbrace{\frac{1}{\kappa}\Big\| \sum_{i < t : c_i = 1} w_i \big(\boldsymbol{\phi}(x_i, a_i) - \boldsymbol{\phi}(x_i, b_i)\big) \Big\|_{\boldsymbol{\Sigma}_t^{-1}}}_{\text{Adversarial term}}.$$

The stochastic term can be upper bounded with the concentration inequality (Lemma D.2). Additionally, by employing our specifically chosen weight, as (4.3), we can control the adversarial term, with $w_i \|\boldsymbol{\phi}(x_i, a_i) - \boldsymbol{\phi}(x_i, b_i)\|_{\boldsymbol{\Sigma}_t^{-1}} \leq \alpha$. Therefore, the adversarial term can be bounded by $\alpha C / \kappa$.

## A.2  REGRET UPPER BOUND

With a similar discussion of the symmetric arm selection rule to Di et al. (2023), the regret defined in (3.2) can be bounded by

$$\text{Regret}(T) \le \sum_{t=1}^{T} \min\left\{4, 2\beta\|\phi(x_t, a_t) - \phi(x_t, b_t)\|_{\boldsymbol{\Sigma}_t^{-1}}\right\}.$$

Note that in our selection of weight $w_t$, it has two possible values. We decompose the summation based on the two cases separately. We have

$$\text{Regret}(T) \le \underbrace{\sum_{w_t=1} \min\left\{4, 2\beta\|\phi(x_t, a_t) - \phi(x_t, b_t)\|_{\boldsymbol{\Sigma}_t^{-1}}\right\}}_{J_1}$$

$$+ \underbrace{\sum_{w_t<1} \min\left\{4, 2\beta\|\phi(x_t, a_t) - \phi(x_t, b_t)\|_{\boldsymbol{\Sigma}_t^{-1}}\right\}}_{J_2}.$$

We consider $J_1, J_2$ separately. For the term $J_1$, we define $\boldsymbol{\Lambda}_t = \lambda\mathbf{I} + \sum_{i \le t-1, w_i=1} \big(\phi(x_i, a_i) - \phi(x_i, b_i)\big)\big(\phi(x_i, a_i) - \phi(x_i, b_i)\big)^{\top}$. Then, we have $\boldsymbol{\Sigma}_t \succeq \boldsymbol{\Lambda}_t$, and therefore

$$\|\phi(x_t, a_t) - \phi(x_t, b_t)\|_{\boldsymbol{\Sigma}_t^{-1}} \le \|\phi(x_t, a_t) - \phi(x_t, b_t)\|_{\boldsymbol{\Lambda}_t^{-1}}.$$

Using Lemma D.3 with $\mathbf{x}_t = \phi(x_t, a_t) - \phi(x_t, b_t)$, we have

$$J_1 \le 4\beta\sqrt{dT\log(1 + 2T/\lambda)}. \tag{A.2}$$

For term $J_2$, we note that $w_t < 1$ implies that $w_t = \alpha/\|\phi(x_t, a_t) - \phi(x_t, b_t)\|_{\boldsymbol{\Sigma}_t^{-1}}$. Therefore, we have

$$J_2 \le \sum_{t=1}^{T} \frac{4\beta}{\alpha} \min\left\{1, \|\sqrt{w_t}(\phi(x_t, a_t) - \phi(x_t, b_t))\|_{\boldsymbol{\Sigma}_t^{-1}}^2\right\}.$$

Using Lemma D.3 with $\mathbf{x}_t' = \sqrt{w_t}(\phi(x_t, a_t) - \phi(x_t, b_t))$, we have

$$J_2 \le \frac{4d\beta\log(1 + 2T/\lambda)}{\alpha}. \tag{A.3}$$

We conclude the proof of regret by combining (A.2) and (A.3).

# B  PROOF OF THEOREMS IN SECTION 5

## B.1  PROOF OF THEOREM 5.3

In this subsection, we provide the proof of Theorem 5.3. We condition on the high-probability event in Lemma 5.1

$$\mathcal{E} = \left\{\|\boldsymbol{\theta}_t - \boldsymbol{\theta}^*\|_{\boldsymbol{\Sigma}_t} \le \beta, \forall t \in [T]\right\}.$$

Let $r_t = 2r^*(x_t, a_t^*) - r^*(x_t, a_t) - r^*(x_t, b_t)$ be the regret incurred in round $t$. The following lemma provides the upper bound of $r_t$.

**Lemma B.1.** Let $0 < \delta < 1$. If we set $\beta = \sqrt{\lambda}B + \big(\alpha C + \sqrt{d\log((1 + 2T/\lambda)/\delta)}\big)/\kappa$, on event $\mathcal{E}$, the regret of Algorithm 1 incurred in round $t$ can be upper bounded by

$$r_t \le \min\left\{4, 2\beta\|\phi(x_t, a_t) - \phi(x_t, b_t)\|_{\boldsymbol{\Sigma}_t^{-1}}\right\}.$$

Moreover, the regret can be upper bounded by

$$\text{Regret}(T) \le \sum_{t=1}^{T} \min\left\{4, 2\beta\|\phi(x_t, a_t) - \phi(x_t, b_t)\|_{\boldsymbol{\Sigma}_t^{-1}}\right\}.$$

With Lemma B.1, we can provide the proof of Theorem 5.3.

*Proof of Theorem 5.3.* Using Lemma B.1, the total regret can be upper bounded by

$$\text{Regret}(T) \leq \sum_{t=1}^{T} \min\left\{4, 2\beta\|\phi(x_t, a_t) - \phi(x_t, b_t)\|_{\mathbf{\Sigma}_t^{-1}}\right\}.$$

Our weight $w_t$ has two possible values. We decompose the summation based on the two cases separately. We have

$$\text{Regret}(T) \leq \underbrace{\sum_{w_t=1} \min\left\{4, 2\beta\|\phi(x_t, a_t) - \phi(x_t, b_t)\|_{\mathbf{\Sigma}_t^{-1}}\right\}}_{J_1}$$

$$+ \underbrace{\sum_{w_t<1} \min\left\{4, 2\beta\|\phi(x_t, a_t) - \phi(x_t, b_t)\|_{\mathbf{\Sigma}_t^{-1}}\right\}}_{J_2}.$$

For the term $J_1$, we consider a partial summation in rounds when $w_t = 1$. Let $\mathbf{\Lambda}_t = \lambda\mathbf{I} + \sum_{i \leq k-1, w_i=1} \left(\phi(x_i, a_i) - \phi(x_i, b_i)\right)\left(\phi(x_i, a_i) - \phi(x_i, b_i)\right)^\top$. Then we have

$$J_1 \leq 4\beta \sum_{t:w_t=1} \min\left\{1, \|\phi(x_t, a_t) - \phi(x_t, b_t)\|_{\mathbf{\Sigma}_t^{-1}}\right\}$$

$$\leq 4\beta \sum_{t:w_t=1} \min\left\{1, \|\phi(x_t, a_t) - \phi(x_t, b_t)\|_{\mathbf{\Lambda}_t^{-1}}\right\}$$

$$\leq 4\beta \sqrt{T \sum_{t:w_t=1} \min\left\{1, \|\phi(x_t, a_t) - \phi(x_t, b_t)\|_{\mathbf{\Lambda}_t^{-1}}^2\right\}}$$

$$\leq 4\beta\sqrt{dT\log(1 + 2T/\lambda)}, \tag{B.1}$$

where the second inequality holds due to $\mathbf{\Sigma}_t \succeq \mathbf{\Lambda}_t$. The third inequality holds due to the Cauchy-Schwartz inequality, The last inequality holds due to Lemma D.3.

For the term $J_2$, the weight in this summation satisfies $w_t < 1$, and therefore $w_t = \alpha/\|\phi(x_t, a_t) - \phi(x_t, b_t)\|_{\mathbf{\Sigma}_t^{-1}}$. Then we have

$$J_2 = \sum_{w_t<1} \min\left\{4, 2\beta\|\phi(x_t, a_t) - \phi(x_t, b_t)\|_{\mathbf{\Sigma}_t^{-1}} w_t \|\phi(x_t, a_t) - \phi(x_t, b_t)\|_{\mathbf{\Sigma}_t^{-1}}/\alpha\right\}$$

$$\leq \sum_{t=1}^{T} \min\left\{4, 2\beta/\alpha\|\sqrt{w_t}(\phi(x_t, a_t) - \phi(x_t, b_t))\|_{\mathbf{\Sigma}_t^{-1}}^2\right\}$$

$$\leq \sum_{t=1}^{T} \frac{4\beta}{\alpha} \min\left\{1, \|\sqrt{w_t}(\phi(x_t, a_t) - \phi(x_t, b_t))\|_{\mathbf{\Sigma}_t^{-1}}^2\right\}$$

$$\leq \frac{4d\beta\log(1 + 2T/\lambda)}{\alpha}, \tag{B.2}$$

where the first equality holds due to the choice of $w_t$. The first inequality holds because each term in the summation is positive. The last inequality holds due to Lemma D.3. Combining (B.1) and (B.2), we complete the proof of Theorem 5.3. $\qquad\square$

## B.2 PROOF OF THEOREM 5.5

*Proof of Theorem 5.5.* Our proof adapts the argument in Bogunovic et al. (2021) to dueling bandits. For any dimension $d$, we construct $d$ instances, each with $\boldsymbol{\theta}_i = \mathbf{e}_i$, where $\mathbf{e}_i$ is the $i$-th standard basis vector. We set the action set $\mathcal{A} = \{\mathbf{e}_i\}_{i=1}^d$. Therefore, in the $i$-th instance, the reward for the $i$-th action will be 1. For the other actions, it will be 0. Therefore, the $i$-th action will be more preferable to any other action. While for other pairs, the feedback is simply a random guess.

Consider an adversary that knows the exact instance. When the comparison involves the $i$-th action, it will corrupt the feedback with a random guess. Otherwise, it will not corrupt. In the $i$-th instance,

the adversary stops the adversarial attack only after $C$ times of comparison involving the $i$-th action. However, after $Cd/4$ rounds, at least $d/2$ actions have not been compared for $C$ times. For the instances corresponding to these actions, the agent learns no information and suffers from $\Omega(dC)$ regret. This completes the proof of Theorem 5.5. □

### B.3 PROOF OF THEOREM 5.7

*Proof of Theorem 5.7.* Here, based on the relationship between $C$ and the threshold $\bar{C}$, we discuss two distinct cases separately.

- In the scenario where $\bar{C} < C$, Algorithm 1 can ensures a trivial regret bound, with the guarantee that $\text{Regret}(T) \leq 2T$.

- In the scenario where $C \leq \bar{C}$, we know that $\bar{C}$ is remains a valid upper bound on the number of adversarial feedback. Under this situation, Algorithm 1 operates successfully with $\bar{C}$ adversarial feedback. Therefore, according to Theorem 5.3, the regret is upper bounded by

$$\text{Regret}(T) \leq \widetilde{O}(d\sqrt{T} + d\bar{C}).$$

□

## C PROOF OF LEMMAS 5.1 AND B.1

### C.1 PROOF OF LEMMA 5.1

*Proof of Lemma 5.1.* Using a similar reasoning in Li et al. (2017), we define some auxiliary quantities

$$G_t(\boldsymbol{\theta}) = \lambda\kappa\boldsymbol{\theta} + \sum_{i=1}^{t-1} w_i \Big[ \sigma\Big( \big(\phi(x_i,a_i) - \phi(x_i,b_i)\big)^\top \boldsymbol{\theta} \Big)$$
$$- \sigma\Big( \big(\phi(x_i,a_i) - \phi(x_i,b_i)\big)^\top \boldsymbol{\theta}^* \Big) \Big] \big(\phi(x_i,a_i) - \phi(x_i,b_i)\big),$$
$$\epsilon_t = l_t - \sigma\Big( \big(\phi(x_t,a_t) - \phi(x_t,b_t)\big)^\top \boldsymbol{\theta}^* \Big),$$
$$\gamma_t = o_t - \sigma\Big( \big(\phi(x_t,a_t) - \phi(x_t,b_t)\big)^\top \boldsymbol{\theta}^* \Big),$$
$$Z_t = \sum_{i=1}^{t-1} w_i \gamma_i \big(\phi(x_i,a_i) - \phi(x_i,b_i)\big).$$

In Algorithm 1, $\boldsymbol{\theta}_t$ is chosen to be the solution to the following equation,

$$\lambda\kappa\boldsymbol{\theta}_t + \sum_{i=1}^{t-1} w_i \Big[ \sigma\Big( \big(\phi(x_i,a_i) - \phi(x_i,b_i)\big)^\top \boldsymbol{\theta}_t \Big) - o_i \Big] \big(\phi(x_i,a_i) - \phi(x_i,b_i)\big) = \mathbf{0}. \quad \text{(C.1)}$$

Then we have

$$G_t(\boldsymbol{\theta}_t) = \lambda\kappa\boldsymbol{\theta}_t + \sum_{i=1}^{t-1} w_i \Big[ \sigma\Big( \big(\phi(x_i,a_i) - \phi(x_i,b_i)\big)^\top \boldsymbol{\theta}_t \Big)$$
$$- \sigma\Big( \big(\phi(x_i,a_i) - \phi(x_i,b_i)\big)^\top \boldsymbol{\theta}^* \Big) \Big] \big(\phi(x_i,a_i) - \phi(x_i,b_i)\big)$$
$$= \sum_{i=1}^{t-1} w_i \Big[ o_i - \sigma\Big( \big(\phi(x_i,a_i) - \phi(x_i,b_i)\big)^\top \boldsymbol{\theta}^* \Big) \Big] \big(\phi(x_i,a_i) - \phi(x_i,b_i)\big)$$
$$= Z_t.$$

The analysis in Li et al. (2017); Di et al. (2023) shows that this equation has a unique solution, with $\boldsymbol{\theta}_t = G_t^{-1}(Z_t)$. Using the mean value theorem, for any $\boldsymbol{\theta}_1, \boldsymbol{\theta}_2 \in \mathbb{R}^d$, there exists $m \in [0,1]$ and $\bar{\boldsymbol{\theta}} = m\boldsymbol{\theta}_1 + (1-m)\boldsymbol{\theta}_2$, such that the following equation holds,

$$G_t(\boldsymbol{\theta}_1) - G_t(\boldsymbol{\theta}_2) = \lambda\kappa(\boldsymbol{\theta}_1 - \boldsymbol{\theta}_2) + \sum_{i=1}^{t-1} w_i \Big[ \sigma\Big( \big(\phi(x_i,a_i) - \phi(x_i,b_i)\big)^\top \boldsymbol{\theta}_1 \Big)$$

$$- \sigma\Big( \big( \phi(x_i, a_i) - \phi(x_i, b_i) \big)^\top \boldsymbol{\theta}_2 \Big) \Big] \big( \phi(x_i, a_i) - \phi(x_i, b_i) \big)$$

$$= \Big[ \lambda \kappa \mathbf{I} + \sum_{i=1}^{t-1} w_i \dot{\sigma}\Big( \big( \phi(x_i, a_i) - \phi(x_i, b_i) \big)^\top \bar{\boldsymbol{\theta}} \Big)$$

$$\big( \phi(x_i, a_i) - \phi(x_i, b_i) \big) \big( \phi(x_i, a_i) - \phi(x_i, b_i) \big)^\top \Big] (\boldsymbol{\theta}_1 - \boldsymbol{\theta}_2).$$

We define $F(\bar{\boldsymbol{\theta}})$ as

$$F(\bar{\boldsymbol{\theta}}) = \lambda \kappa \mathbf{I} + \sum_{i=1}^{t-1} w_i \dot{\sigma}\Big( \big( \phi(x_i, a_i) - \phi(x_i, b_i) \big)^\top \bar{\boldsymbol{\theta}} \Big) \big( \phi(x_i, a_i) - \phi(x_i, b_i) \big) \big( \phi(x_i, a_i) - \phi(x_i, b_i) \big)^\top \Big].$$

Moreover, we can see that $G_t(\boldsymbol{\theta}^*) = \lambda \kappa \boldsymbol{\theta}^*$. Recall $\boldsymbol{\Sigma}_t = \lambda \mathbf{I} + \sum_{i=1}^{t-1} w_i \big( \phi(x_i, a_i) - \phi(x_i, b_i) \big) \big( \phi(x_i, a_i) - \phi(x_i, b_i) \big)^\top$. We have

$$\big\| G_t(\boldsymbol{\theta}_t) - G_t(\boldsymbol{\theta}^*) \big\|^2_{\boldsymbol{\Sigma}_t^{-1}} = (\boldsymbol{\theta}_t - \boldsymbol{\theta}^*)^\top F(\bar{\boldsymbol{\theta}}) \boldsymbol{\Sigma}_t^{-1} F(\bar{\boldsymbol{\theta}})(\boldsymbol{\theta}_t - \boldsymbol{\theta}^*)$$
$$\geq \kappa^2 (\boldsymbol{\theta}_t - \boldsymbol{\theta}^*)^\top \boldsymbol{\Sigma}_t (\boldsymbol{\theta}_t - \boldsymbol{\theta}^*)$$
$$= \kappa^2 \| \boldsymbol{\theta}_t - \boldsymbol{\theta}^* \|^2_{\boldsymbol{\Sigma}_t},$$

where the first inequality holds due to $\dot{\mu}(\cdot) \geq \kappa > 0$ and $F(\bar{\boldsymbol{\theta}}) \succeq \kappa \boldsymbol{\Sigma}_t$. Then we have the following estimate of the estimation error:

$$\| \boldsymbol{\theta}_t - \boldsymbol{\theta}^* \|_{\boldsymbol{\Sigma}_t} \leq \frac{1}{\kappa} \big\| G_t(\boldsymbol{\theta}_t) - G_t(\boldsymbol{\theta}^*) \big\|_{\boldsymbol{\Sigma}_t^{-1}}$$

$$\leq \lambda \| \boldsymbol{\theta}^* \|_{\boldsymbol{\Sigma}_t^{-1}} + \frac{1}{\kappa} \| Z_t \|_{\boldsymbol{\Sigma}_t^{-1}}$$

$$\leq \sqrt{\lambda} \| \boldsymbol{\theta}^* \|_2 + \frac{1}{\kappa} \| Z_t \|_{\boldsymbol{\Sigma}_t^{-1}},$$

where the second inequality holds due to the triangle inequality and $G_t(\boldsymbol{\theta}^*) = \lambda \kappa \boldsymbol{\theta}^*$. The last inequality holds due to $\boldsymbol{\Sigma}_t \succeq \lambda \mathbf{I}$. Finally, we need to bound the $\| Z_t \|_{\boldsymbol{\Sigma}_t^{-1}}$ term. To study the impact of adversarial feedback, we decompose the summation in (A.1) based on the adversarial feedback $c_t$, i.e.,

$$Z_t = \sum_{i<t:c_i=0} w_i \gamma_i \big( \phi(x_i, a_i) - \phi(x_i, b_i) \big) + \sum_{i<t:c_i=1} w_i \gamma_i \big( \phi(x_i, a_i) - \phi(x_i, b_i) \big),$$

When $c_i = 1$, i.e. with adversarial feedback, $|\gamma_i - \epsilon_i| = 1$. On the contrary, when $c_i = 0$, $\gamma_i = \epsilon_i$. Therefore,

$$\sum_{i<t:c_i=0} w_i \gamma_i \big( \phi(x_i, a_i) - \phi(x_i, b_i) \big) = \sum_{i<t:c_i=0} w_i \epsilon_i \big( \phi(x_i, a_i) - \phi(x_i, b_i) \big),$$

$$\sum_{i<t:c_i=1} w_i \gamma_i \big( \phi(x_i, a_i) - \phi(x_i, b_i) \big) = \sum_{i<t:c_i=1} w_i \epsilon_i \big( \phi(x_i, a_i) - \phi(x_i, b_i) \big)$$
$$+ \sum_{i<t:c_i=1} w_i \big( \gamma_i - \epsilon_i \big) \big( \phi(x_i, a_i) - \phi(x_i, b_i) \big).$$

Summing up the two equalties, we have

$$Z_t = \sum_{i=1}^{t-1} w_i \epsilon_i \big( \phi(x_i, a_i) - \phi(x_i, b_i) \big) + \sum_{i<t:c_i=1} w_i \big( \gamma_i - \epsilon_i \big) \big( \phi(x_i, a_i) - \phi(x_i, b_i) \big).$$

Therefore,

$$\| Z_t \|_{\boldsymbol{\Sigma}_t^{-1}} \leq \underbrace{\bigg\| \sum_{i=1}^{t-1} w_i \epsilon_i \big( \phi(x_i, a_i) - \phi(x_i, b_i) \big) \bigg\|_{\boldsymbol{\Sigma}_t^{-1}}}_{I_1} + \underbrace{\bigg\| \sum_{i<t:c_i=1} w_i \big( \phi(x_i, a_i) - \phi(x_i, b_i) \big) \bigg\|_{\boldsymbol{\Sigma}_t^{-1}}}_{I_2}.$$

For the term $I_1$, with probability at least $1 - \delta$, for all $t \in [T]$, it can be bounded by

$$I_1 \leq \sqrt{2 \log \Big( \frac{\det(\boldsymbol{\Sigma}_t)^{1/2} \det(\boldsymbol{\Sigma}_0)^{-1/2}}{\delta} \Big)},$$

due to Lemma D.2. Using $w_i \leq 1$, we have $\sqrt{w_i} \|\phi(x_i, a_i) - \phi(x_i, b_i)\|_2 \leq 2$. Moreover, we have

$$\begin{aligned}
\det(\boldsymbol{\Sigma}_t) &\leq \Big( \frac{\mathrm{Tr}(\boldsymbol{\Sigma}_t)}{d} \Big)^d \\
&= \Big( \frac{d\lambda + \sum_{i=1}^{t-1} w_i \|(\phi(x_i, a_i) - \phi(x_i, b_i))\|_2^2}{d} \Big)^d \\
&\leq \Big( \frac{d\lambda + 2T}{d} \Big)^d,
\end{aligned}$$

where the first inequality holds because for every matrix $\mathbf{A} \in \mathbb{R}^{d \times d}$, $\det \mathbf{A} \leq (\mathrm{Tr}(\mathbf{A})/d)^d$. The second inequality holds due to $\sqrt{w_i} \|\phi(x_i, a_i) - \phi(x_i, b_i)\|_2 \leq 2$. Easy to see that $\det(\boldsymbol{\Sigma}_0) = \lambda^d$. The term $I_1$ can be bounded by

$$I_1 \leq \sqrt{d \log((1 + 2T/\lambda)/\delta)}. \tag{C.2}$$

For $I_2$, with our choice of the weight $w_i$, we have

$$\begin{aligned}
I_2 &\leq \sum_{i < t : c_i = 1} w_i \big\| (\phi(x_i, a_i) - \phi(x_i, b_i)) \big\|_{\boldsymbol{\Sigma}_t^{-1}} \\
&\leq \sum_{i < t : c_i = 1} w_i \big\| (\phi(x_i, a_i) - \phi(x_i, b_i)) \big\|_{\boldsymbol{\Sigma}_i^{-1}} \\
&\leq \sum_{i < t : c_i = 1} \alpha \\
&\leq \alpha C, \tag{C.3}
\end{aligned}$$

where the second inequality holds due to $\boldsymbol{\Sigma}_t \succeq \boldsymbol{\Sigma}_i$. The third inequality holds due to $w_i \leq \alpha / \big\| (\phi(x_i, a_i) - \phi(x_i, b_i)) \big\|_{\boldsymbol{\Sigma}_i^{-1}}$. The last inequality holds due to the definition of $C$. Combining (C.2) and (C.3), we complete the proof of Lemma 5.1. $\qquad\square$

## C.2 Proof of Lemma B.1

*Proof of Lemma B.1.* Let the regret incurred in the $t$-th round by $r_t = 2r^*(x_t, a_t^*) - r^*(x_t, a_t) - r^*(x_t, b_t)$. It can be decomposed as

$$\begin{aligned}
r_t &= 2r^*(x_t, a_t^*) - r^*(x_t, a_t) - r^*(x_t, b_t) \\
&= \langle \phi(x_t, a_t^*) - \phi(x_t, a_t), \boldsymbol{\theta}^* \rangle + \langle \phi(x_t, a_t^*) - \phi(x_t, b_t), \boldsymbol{\theta}^* \rangle \\
&= \langle \phi(x_t, a_t^*) - \phi(x_t, a_t), \boldsymbol{\theta}^* - \boldsymbol{\theta}_t \rangle + \langle \phi(x_t, a_t^*) - \phi(x_t, b_t), \boldsymbol{\theta}^* - \boldsymbol{\theta}_t \rangle \\
&\quad + \langle 2\phi(x_t, a_t^*) - \phi(x_t, a_t) - \phi(x_t, b_t), \boldsymbol{\theta}_t \rangle \\
&\leq \|\phi(x_t, a_t^*) - \phi(x_t, a_t)\|_{\boldsymbol{\Sigma}_t^{-1}} \|\boldsymbol{\theta}^* - \boldsymbol{\theta}_t\|_{\boldsymbol{\Sigma}_t} + \|\phi(x_t, a_t^*) - \phi(x_t, b_t)\|_{\boldsymbol{\Sigma}_t^{-1}} \|\boldsymbol{\theta}^* - \boldsymbol{\theta}_t\|_{\boldsymbol{\Sigma}_t} \\
&\quad + \langle 2\phi(x_t, a_t^*) - \phi(x_t, a_t) - \phi(x_t, b_t), \boldsymbol{\theta}_t \rangle \\
&\leq \beta \|\phi(x_t, a_t^*) - \phi(x_t, a_t)\|_{\boldsymbol{\Sigma}_t^{-1}} + \beta \|\phi(x_t, a_t^*) - \phi(x_t, b_t)\|_{\boldsymbol{\Sigma}_t^{-1}} \\
&\quad + \langle 2\phi(x_t, a_t^*) - \phi(x_t, a_t) - \phi(x_t, b_t), \boldsymbol{\theta}_t \rangle,
\end{aligned}$$

where the first inequality holds due to the Cauchy-Schwarz inequality. The second inequality holds due to the high probability confidence event $\mathcal{E}$. Using our action selection rule, we have

$$\begin{aligned}
\langle \phi(x_t, a_t^*) - \phi(x_t, a_t), \boldsymbol{\theta}_t \rangle + \beta \|\phi(x_t, a_t^*) - \phi(x_t, a_t)\|_{\boldsymbol{\Sigma}_t^{-1}} \\
\leq \langle \phi(x_t, b_t) - \phi(x_t, a_t), \boldsymbol{\theta}_t \rangle + \beta \|\phi(x_t, a_t) - \phi(x_t, b_t)\|_{\boldsymbol{\Sigma}_t^{-1}} \\
\langle \phi(x_t, a_t^*) - \phi(x_t, b_t), \boldsymbol{\theta}_t \rangle + \beta \|\phi(x_t, a_t^*) - \phi(x_t, b_t)\|_{\boldsymbol{\Sigma}_t^{-1}} \\
\leq \langle \phi(x_t, a_t) - \phi(x_t, b_t), \boldsymbol{\theta}_t \rangle + \beta \|\phi(x_t, a_t) - \phi(x_t, b_t)\|_{\boldsymbol{\Sigma}_t^{-1}}.
\end{aligned}$$

Adding the above two inequalities, we have

$$\beta\|\phi(x_t, a_t^*) - \phi(x_t, a_t)\|_{\boldsymbol{\Sigma}_t^{-1}} + \beta\|\phi(x_t, a_t^*) - \phi(x_t, b_t)\|_{\boldsymbol{\Sigma}_t^{-1}}$$

$$\leq \langle \phi(x_t, a_t) + \phi(x_t, b_t) - 2\phi(x_t, a_t^*), \boldsymbol{\theta}_t \rangle + 2\beta\|\phi(x_t, a_t) - \phi(x_t, b_t)\|_{\boldsymbol{\Sigma}_t^{-1}}.$$

Therefore, we prove that the regret in round $t$ can be upper bounded by

$$r_t \leq 2\beta\|\phi(x_t, a_t) - \phi(x_t, b_t)\|_{\boldsymbol{\Sigma}_t^{-1}}.$$

With a simple observation, we have $r_t \leq 4$. Therefore, the total regret can be upper bounded by

$$\text{Regret}(T) \leq \sum_{t=1}^{T} \min\left\{4, 2\beta\|\phi(x_t, a_t) - \phi(x_t, b_t)\|_{\boldsymbol{\Sigma}_t^{-1}}\right\}.$$

$\square$

## D  AUXILIARY LEMMAS

**Lemma D.1** (Azuma–Hoeffding inequality, Cesa-Bianchi & Lugosi 2006). Let $\{\eta_k\}_{k=1}^{K}$ be a martingale difference sequence with respect to a filtration $\{\mathcal{F}_t\}$ satisfying $|\eta_t| \leq R$ for some constant $R$, $\eta_t$ is $\mathcal{F}_{t+1}$-measurable, $\mathbb{E}[\eta_t|\mathcal{F}_t] = 0$. Then for any $0 < \delta < 1$, with probability at least $1 - \delta$, we have

$$\sum_{t=1}^{T} \eta_t \leq R\sqrt{2T \log 1/\delta}.$$

**Lemma D.2** (Lemma 9 Abbasi-Yadkori et al. 2011). Let $\{\epsilon_t\}_{t=1}^{T}$ be a real-valued stochastic process with corresponding filtration $\{\mathcal{F}_t\}_{t=0}^{T}$ such that $\epsilon_t$ is $\mathcal{F}_t$-measurable and $\epsilon_t$ is conditionally $R$-sub-Gaussian, i.e.

$$\forall \lambda \in \mathbb{R}, \mathbb{E}[e^{\lambda \epsilon_t}|\mathcal{F}_{t-1}] \leq \exp\left(\frac{\lambda^2 R^2}{2}\right).$$

Let $\{\mathbf{x}_t\}_{t=1}^{T}$ be an $\mathbb{R}^d$-valued stochastic process where $\mathbf{x}_t$ is $\mathcal{F}_{t-1}$-measurable and for any $t \in [T]$, we further define $\boldsymbol{\Sigma}_t = \lambda\mathbf{I} + \sum_{i=1}^{t} \mathbf{x}_i\mathbf{x}_i^{\top}$. Then with probability at least $1 - \delta$, for all $t \in [T]$, we have

$$\left\|\sum_{i=1}^{T} \mathbf{x}_i\eta_i\right\|_{\boldsymbol{\Sigma}_t^{-1}}^{2} \leq 2R^2 \log\left(\frac{\det(\boldsymbol{\Sigma}_t)^{1/2}\det(\boldsymbol{\Sigma}_0)^{-1/2}}{\delta}\right).$$

**Lemma D.3** (Lemma 11, Abbasi-Yadkori et al. 2011). For any $\lambda > 0$ and sequence $\{\mathbf{x}_t\}_{t=1}^{T} \subseteq \mathbb{R}^d$ for $t \in [T]$, define $\mathbf{Z}_t = \lambda\mathbf{I} + \sum_{i=1}^{t-1} \mathbf{x}_i\mathbf{x}_i^{\top}$. Then, provided that $\|\mathbf{x}_t\|_2 \leq L$ holds for all $t \in [T]$, we have

$$\sum_{t=1}^{T} \min\left\{1, \|\mathbf{x}_t\|_{\mathbf{Z}_t^{-1}}^{2}\right\} \leq 2d\log(1 + TL^2/(d\lambda)).$$

