# OpenReview forum: "Nearly Optimal Algorithms for Contextual Dueling Bandits from Adversarial Feedback"
_ICLR.cc/2025/Conference — Submitted to ICLR 2025_

### Official Review · Reviewer_8zGo · 2024-10-25

**Soundness:** 3
**Presentation:** 3
**Contribution:** 2
**Rating:** 5
**Confidence:** 3

**Summary:**

The authors study contextual dueling bandits under adversarial feedback. They propose an algorithm, RCDB, which utilizes uncertainty-weighted maximum likelihood estimation (MLE). The algorithm achieves a regret of $\tilde{O}(d\sqrt{T} + dC)$, where $C$ is the total number of adversarial feedback instances. They also provide a regret lower bound, showing that the achieved regret bound is nearly optimal. Finally, they conduct numerical experiments to demonstrate the superiority of the proposed algorithm over state-of-the-art dueling bandit algorithms.

**Strengths:**

1. They study dueling bandits under adversarial feedback, which has not been studied before.
2. The achieved regret bound is near optimal.
3. They demonstrate their algorithm using synthetic datasets.

**Weaknesses:**

1. The weighted version for the estimator of dealing with corruption was first proposed in He et al. 2022. Their method is highly rely on this method.

2. For the unknown number of adversarial feedback, it requires the information of upper bound of C.

3. There seems to be a lack of novelty in theoretical analysis.

**Questions:**

1. The main concern is regarding theoretical contribution. Could you explain the nontrivial theoretical contribution?
2. Could you provide a more detailed formulation for $\sigma$?

---

> ### Author Response · Authors · 2024-11-20
> **Response to Reviewer 8zGo**
>
> Thanks for your valuable feedback. We will address your concerns one by one.
>
> **Q1**: The weighted version for the estimator of dealing with corruption was first proposed in [1]. Their method highly relies on this method.
>
> **A1**: While the weighted approach to address corruption was initially introduced in [1], it is limited to linear models. In contrast, our method extends this approach to handle nonlinear models. Detailed intuition behind our approach is provided in Lines 230-299, and a direct comparison with [1] is presented in Lines 299-303. Given the thorough citation of relevant literature and the comprehensive discussion of the relationship with prior work, we do not consider this a limitation of our study.
>
> ----
>
> **Q2**: For the unknown number of adversarial feedback, it requires the information of upper bound of $C$.
>
> **A2**: As outlined in Section 5.2, selecting a tolerance threshold $\bar{C}$ represents a balance between safety and efficiency—a common trade-off in this field. Moreover, fine-tuning hyperparameters during training to achieve desirable empirical results is a well-established standard practice in the field.
>
> In the absence of fine-tuning, a practical approach is to set $\bar{C} = O(\sqrt{T})$. When the actual corruption level is below $\bar{C}$, the regret achieved is optimal. However, if the corruption level exceeds $\bar{C}$, the algorithm will incur linear regret.
> This result is optimal in the following sense: Theorem 4.12 in [1] demonstrates that any linear bandit algorithm achieving an optimal regret upper bound in the uncorrupted setting will fail when the corruption level exceeds $\Omega(\sqrt{T})$, where $T$ is the total number of steps. This observation naturally extends to the domain of dueling bandits. When the corruption level reaches $\bar{C} = \Omega(\sqrt{T})$, any algorithm that performs optimally in the uncorrupted case will suffer linear regret.
>
> In general, it is fundamentally impossible to design an algorithm that adapts to an unknown corruption level while simultaneously maintaining efficient performance guarantees in the uncorrupted setting. This limitation is a well-known challenge when dealing with strong adversaries [1][2]. To further clarify this point, we have included a detailed discussion in Remark 5.8.
>
> ----
>
> **Q3**: There seems to be a lack of novelty in theoretical analysis.
>
> **A3**: As noted by the reviewer, our investigation of dueling bandits under adversarial feedback represents a novel contribution to the field, as this problem has not been previously explored. Given its relevance to real-world applications such as fine-tuning large language models (LLMs) using reinforcement learning from human feedback (RLHF) and the critical importance of developing robust algorithms, achieving optimal results in both uncorrupted and corrupted scenarios is itself a significant and meaningful achievement.
>
> Moreover, we want to emphasize that we study the dueling bandit problem, which is different from the standard linear bandit problem in [1] and incurs several challenges when using uncertainty-based weight in the dueling bandit. As we study the dueling bandit setting, the feedback is binary, given by a preference model $\mathbb{P}(a \succ b| x) = \sigma(r^*(x,a)-r^*(x,b)).$ This divergence in settings leads to a different analysis. Unlike the weighted regression method, our model employs weighted maximum likelihood estimation (MLE) to estimate the underlying parameter $\theta$. The nonlinearity of the function stops us from having a closed-form solution of $\theta$, thus adding difficulty to obtaining the confidence radius. We bypass this difficulty by utilizing an auxilliary vector function $G_{t}({\theta}) = \lambda\kappa\mathrm{\theta} + \sum_{i = 1}^{t-1}w_i\Big[\sigma\big(({\phi}(x_i,a_i)-{\phi}(x_i,b_i))^\top {\theta}\big) -\sigma\big(({\phi}(x_i,a_i)-{\phi}(x_i,b_i))^\top {\theta}^*\big)\Big]\big({\phi}(x_i,a_i)-{\phi}(x_i,b_i)\big)$. The elliptical potential lemma provides an upper bound of $||G_{t}({\theta})||_ {\Sigma_t^{-1}} $. We bridge the gap between this and the confidence radius $||\theta-\theta^*||_{\Sigma_t}$ by mean value theorem (Lines 967 to 978). Therefore, our proof technique is novel.
>
> ----
>
> [1] He et al., Nearly Optimal Algorithms for Linear Contextual Bandits with Adversarial Corruptions, Neurips2022
>
> [2] Ye et al., Corruption-robust algorithms with uncertainty weighting for nonlinear contextual bandits and markov decision processes, ICML2023

---

> ### Author Response · Authors · 2024-11-24
> **Looking forward to your response**
>
> Dear Reviewer 8zGo,
>
> Thank you again for your valuable feedback. In our rebuttal, we believe that we have addressed your concerns about the novelty and the case with unknown corruption level of $C$. As the discussion period is drawing to a close, we wanted to kindly check if you have any additional questions or concerns so that we may address them while there is still time for further discussion.
>
> If our responses have sufficiently addressed your comments, we would sincerely appreciate it if you could consider increasing your score.
>
> Best,
>
> Authors

---

> > ### Comment · Reviewer_8zGo · 2024-11-27
> >
> > Thank you for your response. I acknowledge that your model utilizes weighted maximum likelihood estimation (MLE), which distinguishes it from previous work. However, I still have concerns about the novelty of the algorithm and the theoretical analysis compared to prior research on corruption or dueling (or logistic) bandits. Due to this concern, I will be maintaining my score.

---

> > > ### Author Response · Authors · 2024-11-27
> > > **Thanks for your reply**
> > >
> > > To the best of our knowledge, there are no existing studies on label-flipping adversarial feedback involving a binary Bernoulli variable in dueling bandits (or even logistic bandits). While we build our method upon some techniques from previous works, we believe our study on this unexplored setting is important. Moreover, we believe that our results are very valuable and interesting to this field.
> > >
> > > We would be happy to engage in further discussion if the reviewer could point us to specific works having addressed this problem, or some previous works that can trivially induce our results.
> > >
> > > Looking forward to your reply.

---

> > > > ### Comment · Reviewer_8zGo · 2024-12-01
> > > >
> > > > Thank you for your comments. Your proofs for the confidence bounds of $||\theta_t-\theta^*||$ closely align with the framework of prior work, such as [Li et al., 2017], which uses the (expected) gradient of the log-likelihood (G_t) in its analysis. Similarly, the analysis of adversarial feedback closely follows the framework of [He et al., 2022], leveraging both corruption-independent and corruption-dependent terms (in I_1).  In my view, the technical and algorithmic contributions of this work appear to fall below the expected threshold.

---

### Official Review · Reviewer_WsQg · 2024-10-30

**Soundness:** 3
**Presentation:** 3
**Contribution:** 3
**Rating:** 6
**Confidence:** 3

**Summary:**

The paper studied contextual dueling bandits with adversarial feedback and proposed an uncertainty-weighted MLE algorithm. The authors also provided theoretical upper bound and lower bound for regret. And they also run experiments to verify their findings.

**Strengths:**

It is good to provide both upper bound and lower bound for the problem of contextual dueling bandits with adversarial feedback. And they also provide both theoretical and experimental results.

**Weaknesses:**

See questions for more details.

**Questions:**

1. In Assumption 3.1., you assume the reward is a linear model. Can your method extend to non-linear cases?
2. How to choose the value of the regularization parameter in algorithm 1 specifically?
3. In the experiment setup, you choose the sigmoid function for the preference model. How about the other preference model?

---

> ### Author Response · Authors · 2024-11-20
> **Response to Reviewer WsQg**
>
> Thanks for your positive feedback! We will address your concerns one by one.
>
> **Q1**: In Assumption 3.1., you assume the reward is a linear model. Can your method extend to non-linear cases?
>
> **A1**: The use of a linear reward model with known features is a well-established setting in the literature on dueling bandits, as demonstrated in works such as [1][2][3]. Extending this framework to handle more general nonlinear settings is an exciting and challenging problem. The primary focus of this paper is on designing algorithms that are robust to adversarial feedback. Developing robust algorithms for nonlinear functions falls beyond the scope of this work. We consider this an important direction for future research.
>
> ----
>
> **Q2**: How to choose the value of the regularization parameter in algorithm 1 specifically?
>
> **A2**: In Theorem 5.3, we outline how to select the value of the hyperparameters when $C$ is known. When $C$ is unknown, a practical approach is to begin by choosing a tolerance threshold $\bar{C}$. The choice of $\bar{C}$ reflects a trade-off between safety and efficiency, a common practice in this field.  Moreover, in practical scenarios, hyperparameters can be fine-tuned during the training phase to achieve favorable empirical outcomes, which is a well-established convention in machine learning.
>
> ----
>
> **Q3**: In the experiment setup, you choose the sigmoid function for the preference model. How about the other preference model?
>
> **A3**:
> We choose the sigmoid function because it is the most commonly used choice for modeling preference feedback [1][2][3]. As demonstrated in Assumption 3.2, our theoretical results can work for various kinds of preference models. Thus, the algorithm's performance guarantees remain valid when using other preference models.
>
> ----
>
> [1] Aadirupa Saha. Optimal algorithms for stochastic contextual preference bandits, Neurips2021
>
> [2] Bengs et al., Stochastic contextual dueling bandits under linear stochastic transitivity models, ICML2022
>
> [3] Xiong et al., Gibbs sampling from human feedback: A provable kl-constrained framework for rlhf. ICML2024

---

### Official Review · Reviewer_qjiZ · 2024-11-03

**Soundness:** 3
**Presentation:** 3
**Contribution:** 3
**Rating:** 5
**Confidence:** 4

**Summary:**

This paper introduces a new approach for contextual dueling bandits under adversarial feedback by proposing the Robust Contextual Dueling Bandits (RCDB) algorithm. RCDB incorporates uncertainty-weighted maximum likelihood estimation (MLE) to handle adversarial interference in preference labels, aiming to minimize regret even in adversarial scenarios. The algorithm achieves a near-optimal regret bound that is robust to adversarial feedback, showing both a baseline optimal performance in uncorrupted cases and effective scaling with adversarial interference. Extensive experiments validate RCDB’s superiority over existing dueling bandit algorithms under various adversarial conditions, underscoring its potential for applications that rely on preference-based learning models.

**Strengths:**

+ The problem is timely and important
+ The proposed algorithm achieves (nearly) optimal performance bounds
+ Experiments are also provided

**Weaknesses:**

- Some technical novelty needs to be clarified

**Questions:**

1. It seems to me that the main algorithm is a somewhat extension of the non-dueling counterpart in He et al 2022, i.e., the same uncertainty weight and the same update rule of the weights. Except for the possible non-linearity in dueling (logistic) bandit (which can be handled by existing techniques), it seems to me that the proposed algorithm and its analysis largely follow from prior work.

2. As the dependence of \kappa is a key factor in the logistic bandit, can the authors comment on the optimality of \kappa in the regret upper bound (for both non-corrupted and corrupted terms)?

3. If I understand it correctly, there are two commonly used corruption models. One is the total budget model used in the current paper and the other is the strong corruption model from robust statistics (see Definition 3.1 in [R1). In general, these two models are not comparable. But, given the fact that under the dueling bandit setting, the label is only 0 or 1, it seems to me that these two models are somewhat comparable, especially given the fact the current paper considers the strong adversary model where the adversary can observe the actions. It would be good to see some comments on the comparison of the two commonly used corruption models, i.e., if they are equivalent or one implies the other one. This kind of discussion is useful, since it might make the current results even stronger in the sense that it can be used to talk about some results in another corruption model.



[R1] Zhang, Xuezhou, et al. "Robust policy gradient against strong data corruption." International Conference on Machine Learning. PMLR, 2021.

---

> ### Author Response · Authors · 2024-11-20
> **Response to Reviewer qjiZ**
>
> Thank you for your careful review and detailed feedback. We will address your concerns one by one.
>
> **Q1**: Novelty: It seems to me that the main algorithm is a somewhat extension of the non-dueling counterpart in [1], i.e., the same uncertainty weight and the same update rule of the weights. Except for the possible non-linearity in dueling (logistic) bandit (which can be handled by existing techniques), it seems to me that the proposed algorithm and its analysis largely follow from prior work.
>
> **A1**: We want to emphasize that we study the dueling bandit problem, which is different from the standard linear bandit problem in [1] and incurs several challenges when using uncertainty-based weight in the dueling bandit. As we study the dueling bandit setting, the feedback is binary, given by a preference model $\mathbb{P}(a \succ b| x) = \sigma(r^*(x,a)-r^*(x,b)).$ This divergence in settings leads to a different analysis. Unlike the weighted regression method, our model employs weighted maximum likelihood estimation (MLE) to estimate the underlying parameter $\theta$. The nonlinearity of the function stops us from having a closed-form solution of $\theta$, thus adding difficulty to obtaining the confidence radius. We bypass this difficulty by utilizing an auxilliary vector function $G_{t}({\theta}) = \lambda\kappa\mathrm{\theta} + \sum_{i = 1}^{t-1}w_i\Big[\sigma\big(({\phi}(x_i,a_i)-{\phi}(x_i,b_i))^\top {\theta}\big) -\sigma\big(({\phi}(x_i,a_i)-{\phi}(x_i,b_i))^\top {\theta}^*\big)\Big]\big({\phi}(x_i,a_i)-{\phi}(x_i,b_i)\big)$. The elliptical potential lemma provides an upper bound of $||G_{t}({\theta})||_ {\Sigma_t^{-1}} $. We bridge the gap between this and the confidence radius $||\theta-\theta^*||_{\Sigma_t}$ by mean value theorem (Lines 967 to 978). Therefore, our proof technique is novel.
>
> ----
>
> **Q2**: As the dependence of $\kappa$ is a key factor in the logistic bandit, can the authors comment on the optimality of $\kappa$ in the regret upper bound (for both non-corrupted and corrupted terms)?
>
> **A2**: Thank you for your valuable feedback. In Theorem 5.3, we have demonstrated the dependence of our results on $1/\kappa$. To the best of our knowledge, there is no existing study that examines the optimal dependency of $\kappa$ for dueling bandits in both non-corrupted and corrupted terms. While this is an important issue in logistic bandits, as discussed in the 'Related Works' section (Lines 149-155), it falls outside the scope of our current paper. We will consider this as part of our future work.
>
> ----
>
> **Q3**:  there are two commonly used corruption models. One is the total budget model used in the current paper and the other is the strong corruption model from robust statistics. Are they comparable in the dueling bandit setting?
>
> **A3**: Thank you for your valuable feedback. The key difference between these two models lies in their definitions: the total budget model considers the numerical corruption of the reward, while the strong corruption model focuses on the number of corrupted rounds. In our setting, we consider the label-flipping attack, where the magnitude of adversarial feedback is always 1. Under this condition, the two corruption models are equivalent in the dueling bandit setting. In our revision, we have cited and discussed the mentioned paper to make it more clarified.
>
> ----
>
> [1]: He et al., Nearly Optimal Algorithms for Linear Contextual Bandits with Adversarial Corruptions Neurips2022

---

> ### Author Response · Authors · 2024-11-24
> **Looking forward to your response**
>
> Dear Reviewer qjiZ,
>
> Thank you once again for your valuable and insightful feedback. In our revised paper, we have incorporated a detailed discussion on the two types of corruption models mentioned. In our specific setting with binary feedback, they are indeed equivalent. We are grateful for your remarks, which helped us refine this aspect of our work.
>
> Regarding the novelty concern you raised, we have clarified the technical challenges in our rebuttal. Additionally, we believe our work's novelty lies not only in addressing these challenges but also in the importance of the studied setting. To the best of our knowledge, this is the first theoretical result in contextual dueling bandit with adversarial corruption, and drive a optimal performance within this area.
>
> We are reaching out to see if you have any further concerns or feedback. If our responses have adequately addressed your comments, we would sincerely appreciate it if you could consider increasing your score.
>
> Best,
>
> Authors

---

### Official Review · Reviewer_q8Fc · 2024-11-05

**Soundness:** 3
**Presentation:** 2
**Contribution:** 3
**Rating:** 5
**Confidence:** 3

**Summary:**

This paper studies the contextual dueling bandit problem with adversarial preference feedback in which adversaries may intentionally provide misleading preference feedback. The goal is to design an algorithm that is robust to these adversarial manipulations and has sub-linear regret, i.e., the sum of the difference between two times the maximum reward and total reward of a selected pair of arms.

The authors propose an algorithm named RCDB (Robust Contextual Dueling Bandits) for this problem, which uses uncertainty-dependent weights in the maximum likelihood estimator. They have shown that RCDB achieves a nearly optimal regret bound as long as the number of adversarial feedback, $C=O(\sqrt{T})$. Finally, the authors have shown that RCDB outperforms existing algorithms with various adversarial strategies in different synthetic scenarios, validating its robustness and effectiveness.

**Strengths:**

**The following are the strengths of the paper:**
1. This paper considers a contextual dueling bandit problem with adversarial preference feedback, which is motivated by real-world applications like training LLMs using RLHF.

2. The authors propose a contextual dueling bandi algorithm RCDB that handles adversarial manipulations and then show that RCDB achieves a nearly optimal regret bound.

3. Finally, the authors have demonstrated the different performance aspects (comparing the regret bound of existing dueling bandit algorithms and regret vs the amount of adversarial feedback) of the proposed algorithms on synthetic problem instances.

**Weaknesses:**

**The following are the weaknesses of the paper:**
1. The proposed algorithm RCDB assumes the reward function is linear, which may limit its applicability to real-world settings where the reward functions are non-linear. Even though one can assume a feature map ($\phi$) exists for which reward is a linear function. However, a feature map must be known upfront (Assumption 3.1), which may not always be possible in real-world applications.

2. The performance of RCDB depends on accurately tuning the tolerance threshold $\alpha$ and confidence bound $\beta$, which may vary depending on the number of adversarial feedback. In practice, knowing the amount of adversarial feedback upfront may be impossible (as they are adversarial).

3. It is unclear how to distinguish among the following cases:

    1. Two actions have similar rewards; hence, the preference probability is close to 0.5.
    2. A weak labeler who gives bad quality preference feedback, which may be common in practice.
    3. Adversarial labeler who manipulates the feedback

4. It is not clear why setting weights as defined in Eq. (4.3) is a good approach to dealing with adversarial feedback because it is possible that there is no adversarial feedback in data and still has a high uncertainty.

**Questions:**

Please address the weaknesses of the paper. I have a few more questions/comments:
1. Is the maximum likelihood estimator $\theta_t$ biased due to the small weights of some data without no adversarial manipulation, which is possible as $ ||\phi_i|_ {\Sigma_i^{-1}}$ may be large in the initial rounds.

2. Is there a way to set the weights $(\alpha)$ without knowing the amount of adversarial feedback?


**Minor comment:**
1. Line 248 - Line 293 (expect algorithm description) may not be needed as this is not critical, but it can n create confusion due to the use of Taylor approximation.

2. The authors can check the following paper that considers non-linear reward functions in contextual dueling bandits:
Verma et al., [Neural Dueling Bandits](https://arxiv.org/pdf/2407.17112).

I am open to changing my score based on the authors' responses.

**Details Of Ethics Concerns:**

Since this work is a theoretical paper, I do not find any ethical concerns.

---

> ### Author Response · Authors · 2024-11-20
> **Response to Reviewer q8Fc (Part 1)**
>
> Thanks for your helpful suggestions and comments. We will address your concerns one by one.
>
> **Q1**: the reward function is linear, which may limit its applicability to real-world settings where the reward functions are non-linear. Moreover, the feature map must be known upfront.
>
> **A1**: Using a linear reward model with known features is a well-established framework in the dueling bandits literature, as demonstrated in prior works [1][2][3]. We want to highlight that even in the linear setting, the problem of adversarial feedback by label-flipping has never been fully studied, let alone its extension to general function approximation. Without first establishing a robust and optimal solution for the linear case, generalizing effectively to broader contexts with general reward structures would be highly challenging, if not impossible. Thus, we believe it is important to start with the linear setting as the first step, and the setting we study is significant.
>
> Extending this framework to accommodate more general nonlinear settings is indeed an exciting and challenging direction. After reviewing the paper [4] recommended by the reviewer, we note that it explores approximating the reward model with neural networks, addressing the challenges of nonlinear rewards in dueling bandits. However, the primary focus of our paper is on designing algorithms that are robust to adversarial feedback. Developing robust algorithms for nonlinear functions, such as those modeled with neural networks, falls beyond the current scope of our work. We consider this an important direction for future research. In our revision, we have addressed this limitation and discussed it in the future works section.
>
> ----
>
> **Q2**: Good performance of RCDB requires accurately tuning the tolerance threshold depending on the number of adversarial feedback.
>
> **A2**: As outlined in Section 5.2, a practical approach involves selecting a tolerance threshold $\bar{C}$, which strikes a balance between safety and efficiency---a common trade-off in this domain. Moreover, fine-tuning hyperparameters during training to achieve desirable empirical results is a standard practice in the field. Therefore, we do not view this aspect as an issue.
>
> ----
>
> **Q3**: It is unclear how to distinguish among the following cases:
>
> Two actions have similar rewards; hence, the preference probability is close to 0.5.
>
> A weak labeler who gives bad quality preference feedback, which may be common in practice.
>
> Adversarial labeler who manipulates the feedback
>
> **A3**: Thank you for raising this insightful question. Let us clarify these scenarios further:
>
> 1. **Preference Probability Close to 0.5**      This situation reflects a fundamental challenge faced by all bandit models: decision-making under uncertainty. In our setting, this uncertainty arises from the randomness of the BTL model. When the preference probability is close to 0.5, the comparison is nearly random, meaning the information gain is minimal. However, this is a normal characteristic of the problem and represents the inherent difficulty of learning from uncertain or low-signal comparisons. Addressing this is central to the challenge of dueling bandits.
>
> 2. **Weak Labeler**      In this case, we assume that some of the feedback originates from a reward model $r$ that differs from the true reward $r^*$. This corresponds to a scenario where a labeler has incorrect or inconsistent viewpoints, leading to low-quality feedback.
>
> 3. **Adversarial Labeler**      An adversarial labeler poses a more significant challenge. Unlike a weak labeler, the adversarial labeler can exploit full access to contextual information, actions, and true feedback to craft feedback specifically designed to degrade the algorithm's performance. This scenario includes the weak labeler as a special case when we restrict it to corrupt the data aligned with another reward model $r$. However, adversarial behavior is more general and flexible, resembling strategies such as adversarial attacks or misleading attacks, as discussed in our paper.  Each of these cases has its own distinct mathematical formulation. The setting we study, which includes adversarial labelers, is the most difficult and general of these cases. Consequently, addressing this setting is very significant.

---

> > ### Author Response · Authors · 2024-11-20
> > **Response to Reviewer q8Fc (Part 2)**
> >
> > **Q4**: It is not clear why setting weights as defined in Eq. (4.3) is a good approach to dealing with adversarial feedback because it is possible that there is no adversarial feedback in data and still has a high uncertainty.
> >
> > **A4**: In detail, algorithms find it more challenging to detect corruption in regions of high uncertainty, where adversarial feedback tends to have a greater impact. To address this, our algorithm introduces an uncertainty-dependent weighting scheme for the observed data. This approach helps mitigate potentially large errors in the MLE process.   As a result, the method reduces the regret from $O(Cd\sqrt{T})$ to $O(d\sqrt{T} + dC)$, as shown in Theorem 5.3.
> >
> > As a special case, when there is no adversarial feedback in the data (i.e., $C = 0$), the weights defined in Equation (4.3) and the parameter $\alpha$ in Theorem 5.3 effectively reduce the algorithm to one without weighting. This demonstrates the validity and flexibility of the proposed weighting scheme in Equation (4.3).
> >
> > ----
> >
> > **Q5**: Is the maximum likelihood estimator $\theta_t$ biased due to the small weights of some data without no adversarial manipulation?
> >
> > **A5**: Due to the nonlinearity of the link function, the estimator is inherently biased, regardless of whether weights are applied. Moreover, it is important to note that the uncertainty-based weights $w_i$ are independent of the reward noise $\epsilon_i$ (see Equation (4.3)). Therefore, the weighting mechanism does not introduce any additional effects on the bias of the estimator and we can still establish meaningful theoretical guarantees for our algorithm.
> >
> > ----
> >
> > **Q6**: Is there a way to set the weights $\alpha$ without knowing the amount of adversarial feedback?
> >
> > **A6**: As mentioned in A2, we can start by selecting a tolerance threshold $\bar{C}$ and determine $\alpha$ based on it. In practice, $\alpha$ can be directly treated as a tunable parameter, which typically results in good performance.
> >
> > ----
> >
> > **Q7**: Line 248 - Line 293 (except algorithm description) may not be needed as this is not critical, but it can create confusion due to the use of Taylor approximation.
> >
> > **A7**: While this section may not directly contribute to our proof, we consider it essential because it provides critical insights into the rationale behind our choice of specific weights for the nonlinear model. Moreover, the principles discussed in this section could be applicable to similar challenges involving uncertainty quantification in various contexts, making the analysis potentially valuable beyond its immediate scope.
> >
> > ----
> >
> > [1] Aadirupa Saha. Optimal algorithms for stochastic contextual preference bandits, Neurips2021
> >
> > [2] Bengs et al., Stochastic contextual dueling bandits under linear stochastic transitivity models, ICML2022
> >
> > [3] Xiong et al., Gibbs sampling from human feedback: A provable kl-constrained framework for rlhf. ICML2024
> >
> > [4] Vermal et al., 2024 Neural dueling bandits

---

> > > ### Comment · Reviewer_q8Fc · 2024-12-02
> > >
> > > Dear Authors,
> > >
> > > \
> > > Thank you for your detailed response. As mentioned by Reviewer 8zGo, the overall technical and algorithmic contributions of this work appear to fall below the expected threshold, having an efficient way to choose the value of tolerance threshold $\bar{C}$ for any problem with $C = o(T)$ while achieving the best performance (i.e., lowest regret) can be a novel contribution.
> > >
> > > I believe A2 means Section A.2 in the Appendix. I checked A.2, but I did not get a clear answer to my question, i.e., how do we select the value of tolerance threshold $\bar{C}$ when the amount of adversarial feedback is unknown? Since we may not know how much the amount of adversarial feedback in many real-life applications and having proper weights (hence choosing the right value of tolerance threshold) is core to the algorithm’s performance, choosing a wrong value can lead to worse regret bounds (choosing too large value) or even linear regret bound (choosing too small value). Because of this, I tend to maintain my score.

---

> > > > ### Author Response · Authors · 2024-12-03
> > > > **Thanks for your reply**
> > > >
> > > > Thanks for your response and comments. For clarification, our A2 in A6 means Answer 2(response to Question 2(Q2)), not Section A.2 . We want to clarify that the ideal case mentioned by the reviewer is **impossible**. As we study the strong adversarial case (we guess that it's where this confusion comes from), our Section 5.2 (especially Remark 5.6) shows that when the corruption $C=\Omega(\sqrt{T})$, for instance, $C = O(T^{3/4})$, **any** algorithm cannot simultaneously achieve near-optimal regret when uncorrupted and maintain sublinear regret with corruption. This includes dynamically choosing the $\bar C$ in our algorithm. Thus our results have been the best possible one in our setting.
> > > >
> > > > We are happy to receive your feedback if you have any further concerns.

---

> ### Author Response · Authors · 2024-11-24
> **Looking forward to your response**
>
> Dear Reviewer q8Fc,
>
> Thank you once again for your valuable feedback. In our rebuttal, we have carefully addressed the concerns you raised and provided clarifications where needed. Regarding the problem of linear rewards, we have thoroughly reviewed the paper you suggested and think that it can be an interesting future extension of our work.  We have included a discussion of its relevance to our work in the revised manuscript.
>
> As the discussion period is close to the conclusion, we want to kindly ask if you have any additional concerns or questions. If our responses have sufficiently addressed your feedback, we would greatly appreciate it if you could consider increasing the score.
>
> Best,
>
> Authors

---

### Meta-Review · Area_Chair_9VYp · 2024-12-22

**Metareview:**

This paper examines the contextual dueling bandit problem with potentially adversarial feedback and proposes a robust algorithm. The algorithm is shown to achieve a regret upper bound that depends on the total amount of adversarial feedback, $C$. The main weaknesses include the requirement of knowing $C$ in advance, the limited technical novelty, and the restriction of the approach to linear models.

I do not consider the limitation to linear models to be a significant issue in itself. However, since the primary motivation of the paper is the application to training large language models (LLMs) via RLHF, it is essential to provide a justification for the use of linear models in this context, which is insufficiently addressed in the paper.

The reviewers’ evaluations are borderline, but the above concerns remain unresolved and lean slightly negative. Therefore, I cannot support the acceptance of this paper.

**Additional Comments On Reviewer Discussion:**

The reviewers raised concerns regarding the practicality of requiring prior knowledge of the total amount of adversarial feedback, $C$, and the limited technical novelty of the work. While the authors provided counterarguments, these were not sufficient to fully convince the reviewers to the extent of overturning their initial evaluations.

---

### Decision · Program_Chairs · 2025-01-22

Reject